# Single-cell multiomics reveals the interplay of clonal evolution and cellular plasticity in hepatoblastoma

Amélie Roehrig[1], Theo Z. Hirsch [1], Aurore Pire [1], Guillaume Morcrette[1,2], Barkha Gupta[1], Charles Marcaillou[3], Sandrine Imbeaud [1], Christophe Chardot[4], Emmanuel Gonzales[5], Emmanuel Jacquemin[5], Masahiro Sekiguchi [6], Junko Takita [6,7], Genta Nagae [8], Eiso Hiyama [9,10], Florent Guérin [11], Monique Fabre[12], Isabelle Aerts[13], Sophie Taque[14], Véronique Laithier[15], Sophie Branchereau[11], Catherine Guettier[16], Laurence Brugières[17], Brice Fresneau [17], Jessica Zucman-Rossi [1,18,21] ✉ & Eric Letouzé [1,19,20,21] ✉

Hepatoblastomas (HB) display heterogeneous cellular phenotypes that influence the clinical outcome, but the underlying mechanisms are poorly understood. Here, we use a single-cell multiomic strategy to unravel the molecular determinants of this plasticity. We identify a continuum of HB cell states between hepatocytic (scH), liver progenitor (scLP) and mesenchymal (scM) differentiation poles, with an intermediate scH/LP population bordering scLP and scH areas in spatial transcriptomics. Chromatin accessibility landscapes reveal the gene regulatory networks of each differentiation pole, and the sequence of transcription factor activations underlying cell state transitions. Single-cell mapping of somatic alterations reveals the clonal architecture of each tumor, showing that each genetic subclone displays its own range of cellular plasticity across differentiation states. The most scLP subclones, overexpressing stem cell and DNA repair genes, proliferate faster after neo-adjuvant chemotherapy. These results highlight how the interplay of clonal evolution and epigenetic plasticity shapes the potential of HB subclones to respond to chemotherapy.

Hepatoblastoma (HB), usually diagnosed during the first 5 years of life, is the most frequent pediatric liver tumor. It accounts for ~1% of pediatric cancers and its incidence is rising[1,2]. HB treatment relies on neo-adjuvant cisplatin-based chemotherapy followed by surgery, resulting in a 5-year survival rate near 80%. However, chemo-resistant HB are confronted to a lack of therapeutic alternatives and thus lead to poor prognosis[3,4].

The genomic landscape of HB is relatively simple, with ß-catenin activating mutations in almost all the tumors, alterations of the 11p15.5 imprinted locus in ~85% of cases, and a dozen of other drivers altered at low frequency including *TERT, NFE2L2, ARID1A, RPS6KA3, MDM4* or *CCND1*[5,6]. By contrast, HB are phenotypically heterogeneous, with 3

main histological patterns - fetal, embryonal, and mesenchymal - that often coexist within a single tumor[7]. Bulk transcriptomic studies[5,8–11] identified 3 major groups related to histological subtypes. 'Hepatocytic' (H) samples are well-differentiated with fetal histology. 'Liver Progenitor' (LP) samples are less differentiated, more proliferative and associated with embryonal histology. 'Mesenchymal' (M) samples lack liver differentiation features and display mesenchymal cell morphologies. Transcriptomic subgroups display striking spatial and longitudinal heterogeneity[5], extending the heterogeneity described at the histological level, which reflects the ability of tumor cells to change their phenotype. This plasticity, confirmed by us[5] and others[12,13] in

single-cell RNA-seq studies, is particularly important to understand response to treatment since the LP phenotype has been associated with chemoresistance[5]. However, the molecular mechanisms underlying the plasticity of HB cells remain unknown. In particular, the relative roles of genetic evolution and epigenetic plasticity in shaping HB cell phenotypes has not been investigated.

In this work, we integrate whole genome sequencing (WGS) and single-nucleus Multiome, allowing simultaneous profiling of gene expression (RNA-seq) and open chromatin (ATAC-seq) from the same cells, to reconstruct the genetic, transcriptomic and epigenomic evolution of 6 representative HB (~23,000 cells). We explore the diversity of cell states, reconstruct the gene regulatory networks and chromatin changes regulating phenotypic switches, and study the interplay of clonal evolution and cellular plasticity.

## Results

### Single-cell multiomic characterization of hepatoblastoma

We selected 6 HB samples from the Hirsch data set[5], representative of HB diversity based on their bulk RNA-seq profiles (Fig. 1a and Supplementary Data 1). This series includes samples of the Hepatocytic (H, $n = 2$), Liver Progenitor (LP, $n = 3$) and Mesenchymal (M, $n = 1$) transcriptomic subgroups (Supplementary Fig. 1). Two synchronous samples correspond to LP (#2959 T) and H (#2960 T) regions of the same tumor. Histological annotations of the mirror blocks (Supplementary Fig. 2) revealed various contributions of fetal, embryonal and mesenchymal (including immature mesenchymal and osteoid) components. Thus, this sample selection covers the molecular and histological diversity of HB cells, with both inter- and intra-sample heterogeneity. All patients received cisplatin-based neoadjuvant chemotherapy before surgery and sampling, with good response except one patient (#3662 T) who displayed a re-increase of serum alphafetoprotein (AFP) during treatment. All patients are in complete remission (median follow-up = 9.5 years), except #3133 T who relapsed and died 21 months post-diagnosis (Supplementary Fig. 3). The 6 HB samples were analyzed by bulk whole genome sequencing and single-nucleus (sn) Multiome (RNA-seq + ATAC-seq). We also analyzed 2 matched non-tumor liver samples, one of which (#3377 N) showed pre-neoplastic colonization by cells with mosaic alteration of 11p15 locus[5]. Driver alterations included activating *CTNNB1* mutations in all samples, copy-neutral loss of heterozygosity (cnLOH) of 11p15 (*IGF2* locus) in 5/6 samples, and private alterations of *TERT*, *APC*, *ARID1A*, *DDX3X* and *IRF2* (Fig. 1b). After quality controls (see Methods and Supplementary Fig. 4), we obtained 21,150 snRNA-seq profiles (median 8408 UMIs and 3263 genes per cell) and 17,649 snATAC-seq profiles (median 10,027 fragments per cell with TSS enrichment = 9.68), with an overlap of 15,832 cells (Supplementary Data 2). Unsupervised classifications based on snRNA-seq and snATAC-seq profiles revealed non-tumor cell clusters grouped by cell type, and tumor cell clusters grouped by sample of origin (Fig. 1c). Single-cell virtual copy-number profiles revealed inter-tumor heterogeneity, highly consistent with WGS data (Supplementary Fig. 5), explaining at least in part the sample-wise clustering of tumor cells. Using a dedicated approach based on germline polymorphisms (see "Methods"), we also identified the cnLOH of 11p15 in single cells of 5/6 samples (Fig. 1d). Some of the somatic mutations identified by WGS were also detected in single cell data, including the expected activating *CTNNB1* mutations in each sample (Fig. 1e). We used virtual copy-number profiles to identify tumor cells (Fig. 1f), and established marker genes to annotate immune cell clusters, with a predominance of T cells and macrophages (Fig. 1c). The median tumor cell content in the 6 HB samples was 90% (range 65%-96%). Liver Progenitor samples were devoid of immune cells compared to Hepatocytic samples (12% vs. 32% on average, Fig. 1g), consistent with their immune-cold signature in bulk RNA-seq[5]. We next leveraged this integrated data set with genomic, single-cell transcriptomic and epigenomic profiles to explore the determinants of HB cell heterogeneity.

### Hepatoblastoma cells display continuous states along two differentiation axes

Uniform Manifold Approximation and Projection (UMAP) and t-distributed stochastic neighbor embedding (t-SNE) are widely used to explore cellular heterogeneity in single-cell data. Yet, when applied to cancer data, both methods tend to be highly sensitive to copy-number and group tumor cells by sample of origin rather than molecular group. This was also the case in our series (Fig. 1c). Integration methods (like Harmony[14] or scVi[15]) are well suited to integrate normal cells, expected to have similar transcriptomes across samples. By contrast, tumor cells display important inter-sample biological variation, notably due to tumor-specific driver alterations and copy-number changes. Principal component analysis (PCA) was previously used to separate biological from technical and tumor-specific variations in scRNA-seq data, and to identify shared transcriptional programmes in tumor cells[16]. Applied to our 14,448 tumor cells, PCA revealed 2 main axes of variation, highly correlated with cell differentiation markers (Fig. 2a–c). Principal component 1 (PC1) was correlated with the expression of M markers (cor = 0.97, $P < 2.2E{-}16$); PC2 was correlated with the expression of LP markers (cor=0.82, $P < 2.2E{-}16$) and negatively correlated with the expression of H markers (cor = −0.88, $P < 2.2E{-}16$). Tumor cells of the M sample #3610 T defined a separate cluster with high PC1 activity. By contrast, tumor cells of the LP and H samples were intermingled and continuously distributed along PC2. Similarly, individual UMAP classifications did not reveal distinct H / LP clusters but a gradient of H / LP differentiation within each tumor (Supplementary Fig. 6). In an independent scRNA-seq data set comprising 6244 tumor cells from 9 HB[13], principal component analysis also distributed cells along two main axes related to H, LP and M differentiation (Supplementary Fig. 7). Applying thresholds on PC1 and PC2, we defined 4 single-cell states: scH, scLP and scM corresponding to the 3 poles of differentiation, and scH/LP corresponding to intermediate cells expressing both H and LP markers at moderate levels (Fig. 2d). Chromatin accessibility was closely correlated with expression levels, with intermediate accessibility of both scH and scLP markers in scH/LP cells (Fig. 2e). Single-cell states were present in each sample in various proportions consistent with bulk RNA-seq classification of matched samples (Fig. 2f), and co-localized in snRNA-seq and snATAC-seq UMAPs within each sample-wise cluster (Fig. 2g, h). To visualize the spatial organization of cell states, we generated Visium spatial gene expression data for sample #3133 T. Deconvolution of single cell states in Visium spots was strongly associated with their histological annotation (Fig. 3a,b and Supplementary Fig. 8). Globally, we observed a spatial clustering of cells expressing scLP or scH markers (Supplementary Data 3), corresponding to areas of embryonal and fetal histology, respectively (Fig. 3a, b). Yet, expression changes were gradual, with an anti-correlation of scLP and scH markers. In particular, we identified spots displaying moderate expression of both LP and H markers, corresponding to the intermediate scH/LP state, located at the interface between fetal and embryonal areas (Fig. 3c–g). Altogether, these findings indicate a plasticity of HB cells between 3 differentiation poles (H, LP, M), with a continuum of intermediate states between the LP and H poles.

### Chromatin accessibility landscapes of HB differentiation poles

We used snATAC-seq data to characterize the chromatin accessibility landscape of HB cells. In total, 158,707 ATAC-seq peaks were identified in the data set, 21,627 of which displayed significantly different accessibility between tumor and non-tumor liver cells ( | log₂FC| ≥ 1 & FDR ≤ 10⁻³, Supplementary Data 4 and 5). We notably examined peaks located in imprinted regions implicated in HB development (Supplementary Data 6 and Supplementary Fig. 9). Tumor cells displayed increased accessibility of 52 peaks in the 11p15 region, 8 of which linked with the expression of *IGF2* oncogene, and 32 peaks in the *DLK1/MEG3* locus at 14q32, including peaks linked with the expression of several

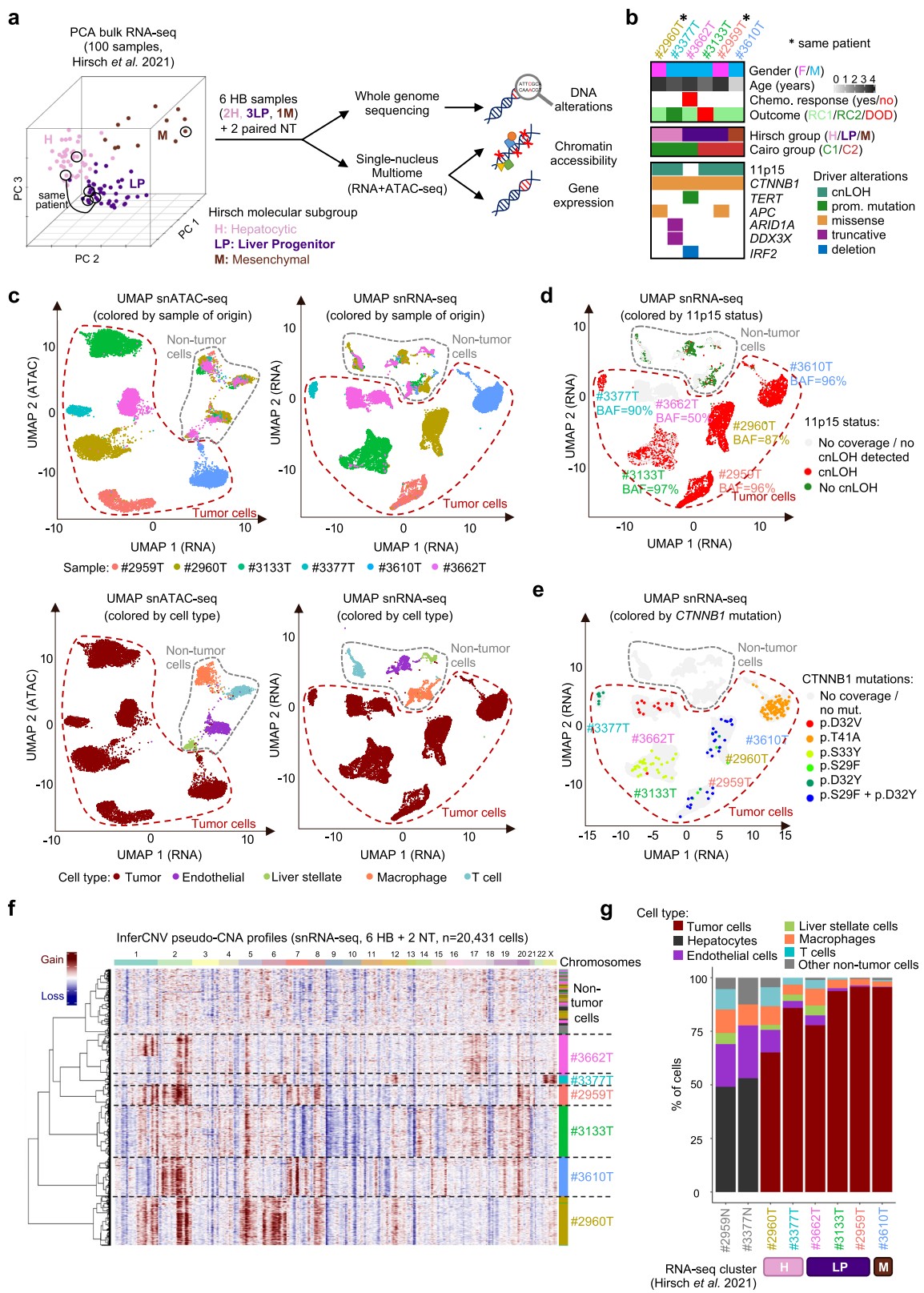

genes belonging to the bad prognosis "14q32 signature" (*DLK1*, *MEG3*, *RTL1*, *MEG8* and *MEG9*)[17]. 57,324 peaks displayed significantly different accessibility between the scH, scLP and scM poles (Fig. 4a and Supplementary Data 4 and 5). ScM cells displayed the most divergent chromatin accessibility profiles, with 48,846 differential peaks as compared with epithelial tumor cells (union of scH, scLP and scH/LP states). Epithelial cells displayed increased accessibility of liver-specific

enhancers (EnhA1, EnhG2) marked by H3K27Ac in ROADMAP adult normal liver[18]. By contrast, scM cells displayed increased accessibility of bivalent transcription start sites (TSS) and enhancers (TssBiv, EnhBiv), and regions repressed by Polycomb (ReprPC, ReprPCWk), all of which bear the repressive mark H3K27Me3 in normal liver (Fig. 4b). ScLP and scH cells had more similar chromatin accessibility profiles, with only 14,625 differential peaks. Peaks more accessible in scH cells

**Fig. 1 | Single-cell multiomic characterization of hepatoblastoma. a** Principal component analysis (PCA) of 100 HB with bulk RNA-seq data. Tumors are colored according to their molecular group as assigned by Hirsch et al.[5]. Black circles indicate the 6 representative HB samples selected for single-cell Multiome and whole genome sequencing (WGS) in this study. **b** Main clinical and molecular annotations of the 6 HB samples. **c** Uniform manifold approximation and projection (UMAP) of all cells from the 6 HB based on their snATAC-seq (left) and snRNA-seq (right) profiles, annotated by sample of origin (top) or cell type (bottom). **d** Projection of 11p15 copy-neutral LOH on the snRNA-seq UMAP. We used germline SNPs to identify in single cells the cnLOH events detected in WGS data (see

Methods). We also computed a B Allele Frequency (BAF) at the sample level corresponding to the proportion of paternal alleles in the LOH region over all cells. A BAF of 50% is expected in absence of LOH. **e** Projection of *CTNNB1* mutations on the snRNA-seq UMAP. Mutations identified in WGS were detected in single cells using *scReadCounts* (see Methods). Only mutations detected in ≥3 cells are shown for each sample. **f** Virtual copy-number alteration (CNA) profiles discriminate tumor and non-tumor cells, cluster cells by sample of origin and reveal intra-sample heterogeneity. **g** Proportion of tumor and non-tumor cell types in each sample. The molecular group of the matched bulk RNA-seq sample is indicated below. Source data are provided in the Source Data file.

---

were enriched in liver-specific enhancers and ZNF repeats, whereas peaks more accessible in scLP cells were enriched in bivalent and Polycomb-repressed regions marked by H3K27Me3 in normal liver (Fig. 4c). These changes, correlated with previously identified methylation components of H, LP and M molecular groups (Supplementary Fig. 10), reflect the level of pluripotency and engagement towards hepatocytic differentiation of each cell state. Motif enrichment analysis revealed transcription factors (TFs) whose binding motifs are more accessible in each differentiation state. TFs with increased motif accessibility in scM cells included the nuclear receptors NR5A1 and ESR1, LEF1 involved in Wnt signaling pathway, and several TFs involved in embryonic development like TWIST1 or ZIC2. By contrast, epithelial HB cells displayed increased accessibility of the liver differentiation TFs HNF1A/4A motifs. Among epithelial cells, scH displayed opening of the hepatocyte TFs CEBPB/D and AR motifs, whereas scLP displayed increased motif accessibility of the Myc-associated zinc finger protein (MAZ), LHX1 involved in embryonic development, and SOX TFs associated with the maintenance of a progenitor phenotype[19].

### Progressive activation of gene regulatory networks underlying cell state transitions

In order to robustly define the gene regulatory networks (GRNs), i.e. the key TFs and target genes defining each cellular state, we integrated single-cell transcriptome and chromatin accessibility profiles of the 14,448 HB cells, together with bulk RNA-seq from 100 HB[5]. We first selected the top 20 TFs associated with each cell state according to a composite score integrating up-regulation in single-nucleus and bulk RNA-seq together with motif accessibility in snATAC-seq data (see Methods section). For some of these TFs, ChIP-seq data was available in the ReMAP database[20] and we could demonstrate a significant overlap between their CHIP-seq peaks and the ATAC-seq peaks with increased accessibility in the associated cell state (Fig. 4d, Supplementary Data 7 and Supplementary Fig. 11). We then identified the targets of each TF as correlated genes with a chromatin accessibility peak containing the TF motif. Finally, we clustered TFs and target genes according to their expression correlation score in single-cells (Fig. 5a). We identified 4 TF-target modules (Supplementary Data 7 and 8), more or less active in each cell state (Fig. 5b). To ensure the robustness of the GRNs, we validated the correlations of TFs and their targets in 6244 tumor cells from 9 HB[13], and in a compendium of 314 bulk RNA-seq profiles from 5 published series[5,6,10,11,17] (Supplementary Fig. 12a). The scM module (22 TFs, 2244 targets) included, in addition to the already mentioned ZIC and TWIST families, several TFs involved in embryonic development like TBX4/5, WT1, SOX8 or ALX3. The sc-epi module (14 TFs, 2690 targets) comprised the hepatocyte nuclear factors HNF1A/4A/4G and others essential for liver development (GATA4, ONECUT1/2, or FOXA1/3). It was active in all epithelial cells, and particularly in scH/LP cells. By contrast, the activity of the scH module (14 TFs, 1647 targets) increased gradually between scLP, scH/LP and scH cells (Fig. 5b). This module comprised TFs involved in the differentiation of hepatocytes (CEBPB/D[21], THRB[22]) and regulating specific functions like xenobiotic metabolism (AHR[23]) or bile acid synthesis (NR1H4[24]). Finally, the scLP module (11 TFs, 2038 targets) included MAZ, LHX1,

SOX4/12, MYCN and TFs regulating cell proliferation (HMGAD, E2F5). While the scM module was specifically active in scM cells, the borders between sc-LP, sc-epi and sc-H modules were porous. We further explored the timing of TF activation along the LP-H differentiation axis by ordering cells and TFs according to the snRNA-seq component PC2 (Fig. 6). Some TFs were activated at the extremities of the differentiation spectrum, like LHX1 in scLP, or CEBPB/D in scH cells. Others like SOX4 (scLP) or AHR (scH) were active in a broader range of differentiation states. Finally, TFs of the sc-epi module (HNF1A, GATA4, FOXA3, ONECUT1/2) displayed maximal activation in scH/LP cells, at intermediate stages of differentiation. This ordering, validated in bulk RNA-seq data (Supplementary Fig. 12b), reflects the progressive activation of GRNs allowing HB cells to switch between scLP and scH phenotypes.

### Deconvolution of HB cell states in bulk RNA-seq data

To further validate the associations of HB cell states with histological components and GRN activity, we used Bisque deconvolution tool[25] to estimate the proportions of stromal cells and HB cell states in our large cohort of 100 bulk RNA-seq profiles[5]. Cell proportions estimated from bulk RNA-seq were consistent with those found in matched single-nucleus data for the 6 samples analyzed in this study (Fig. 7a). In the whole cohort, cell state proportions were strongly associated with histological annotations and bulk molecular groups (Fig. 7b). However, tumor samples were rarely pure and rather comprised a mixture of cell states. In particular, most samples of the LP group displayed various contributions of scH cells, in agreement with the identification of fetal histological components in the mirror blocks. Thus, deconvolution of single-cell states may provide a better assessment of intra-sample heterogeneity than bulk molecular groups. Finally, cell state proportions were significantly correlated with the expression of their GRNs (Fig. 7c). Similar analyses confirmed these associations in 214 other HB from 4 published studies[6,10,11,17] (Supplementary Fig. 13), with comparable cell state proportions between pre- and post-chemotherapy samples (Supplementary Fig. 14).

### Mapping genetic subclones in single-cell data

Tumor progression involves successive rounds of clonal expansions, in which tumor cells with a selective advantage proliferate faster than their neighbors and generate subclones. These subclones can be tracked by the genetic alterations - copy-number alterations (CNAs) and mutations - inherited from the common ancestor cell. To reconstruct the clonal architecture at single-cell scale, we first leveraged the high-quality somatic mutations identified in matched WGS data. Overall, 347/11,798 (2.9%) somatic mutations identified by WGS were detected in single-cell data. 9.7% of cells displayed at least one somatic mutation, allowing to assign cells to their sample of origin with 97% precision (Supplementary Fig. 15). In patient #2959 (2 synchronous HB samples), WGS revealed 382 trunk mutations common to both samples, 534 subclone 1 (cl1) mutations specific to sample #2960 T, and 499 subclone 2 (cl2) mutations specific to sample #2959 T (Fig. 8a). We used scReadCounts to map these mutations in single-cell data (Fig. 8b). As expected, trunk mutations (n = 34 detected in single cells) were encountered in all snRNA-seq clusters. By contrast, cl1 (n = 25) and cl2

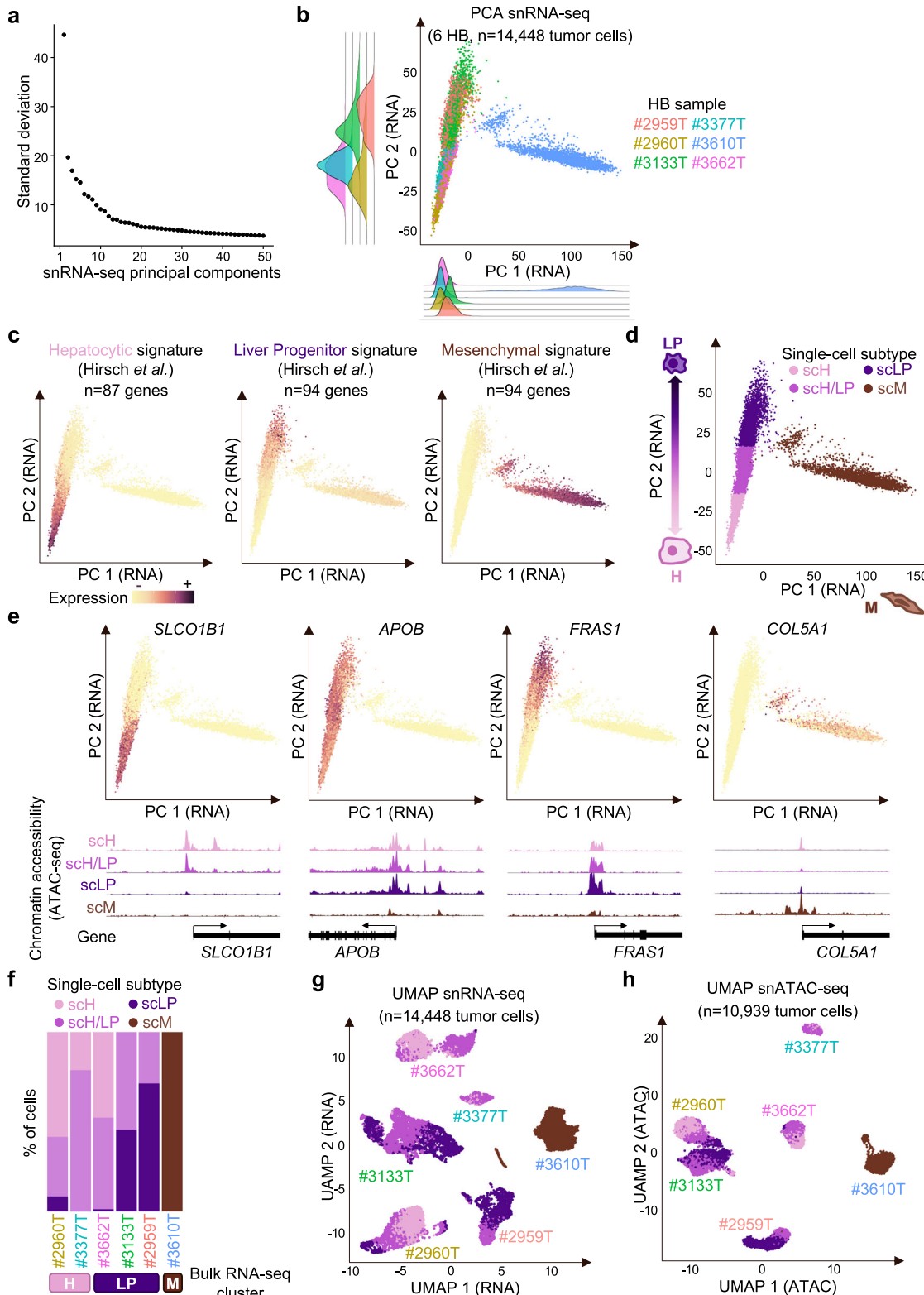

**Fig. 2 | Hepatoblastoma cells display a continuum of states between 3 differentiation poles. a** Scree plot showing the standard deviation of principal components (PCs) in the snRNA-seq data of tumor cells. **b** Projection of tumor cells from the 6 HB samples on the 2 first principal components. Tumor cells are colored by sample. Density plots show the distribution of cells from each sample along the two PCA axes. **c** Mean expression of Hepatocytic, Liver Progenitor and Mesenchymal signatures from bulk RNA-seq[5] are shown onto the snRNA-seq PCA of tumor cells. **d** Single-cell states defined using thresholds on PC1 and PC2 include 3 poles (scH, scLP and scM) and one intermediate state (scH/LP). **e** Expression of representative markers of single-cell states (top). *SLCO1B1*, *FRAS1* and *COL5A1* are markers of the scH, scLP and scM state, respectively. *APOB* is expressed in all epithelial cells (scH, scH/LP and scLP). ATAC-seq tracks below indicate chromatin accessibility at the gene promoter in each cell state. **f** Proportion of single-cell states in each tumor. The molecular group of the matched bulk RNA-seq sample is indicated below. **g** snRNA-seq UMAP of all tumor cells colored by single-cell state. **h** snATAC-seq UMAP of all tumor cells (that passed snRNA-seq and snATAC-seq QC) colored by single-cell state. Source data are provided in the Source Data file.

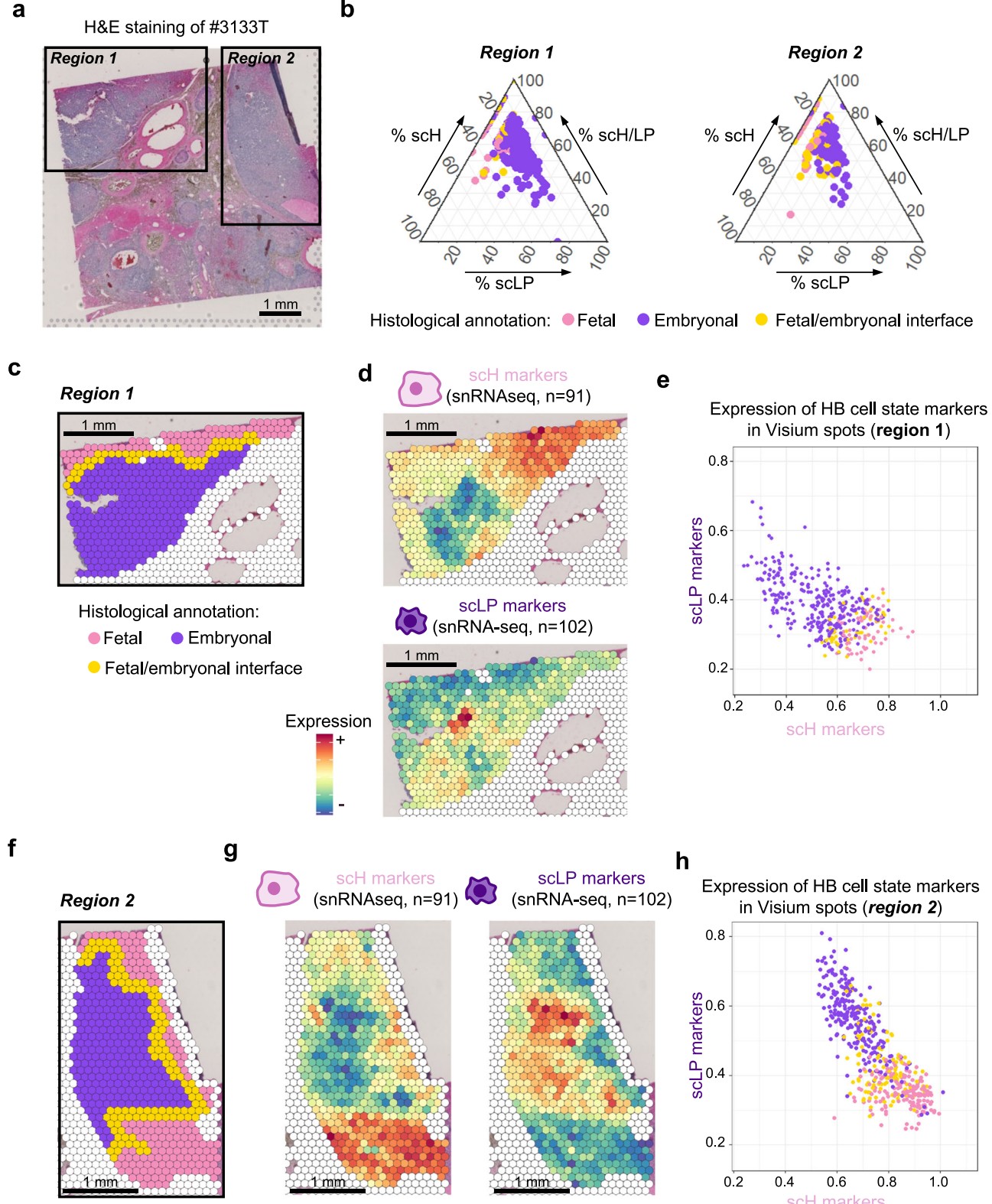

**Fig. 3 | Spatial transcriptomics analysis of tumor #3133 T. a** Hematoxylin and eosin (H&E) staining in the formalin-fixed paraffin-embedded (FFPE) section of one tumor (#3133 T). The two annotated regions correspond to embryonal nodules surrounded by fetal areas. **b** Ternary graph showing the deconvoluted proportions of epithelial HB cell states in each Visium spot of regions 1 (left) and 2 (right). Visium spots were annotated by a pathologist and are colored by histology. **c** Focus on region 1 of the FFPE slide with a nodule of embryonal cells surrounded by fetal cells. **d** Spatial distribution of the mean expression of scH and scLP markers (Supplementary Data 3) in region 1. **e** Anti-correlation of scH and scLP markers across Visium spots in region 1, colored according to their histological annotation in (**c**). **f** Focus on region 2 of the FFPE slide, with spots annotated by histology. **g** Spatial distribution of the mean expression of scH and scLP markers in region 2. **h** Anti-correlation of scH and scLP markers across Visium spots in region 2, colored according to their histological annotation in panel **f**. Source data are provided in the Source Data file.

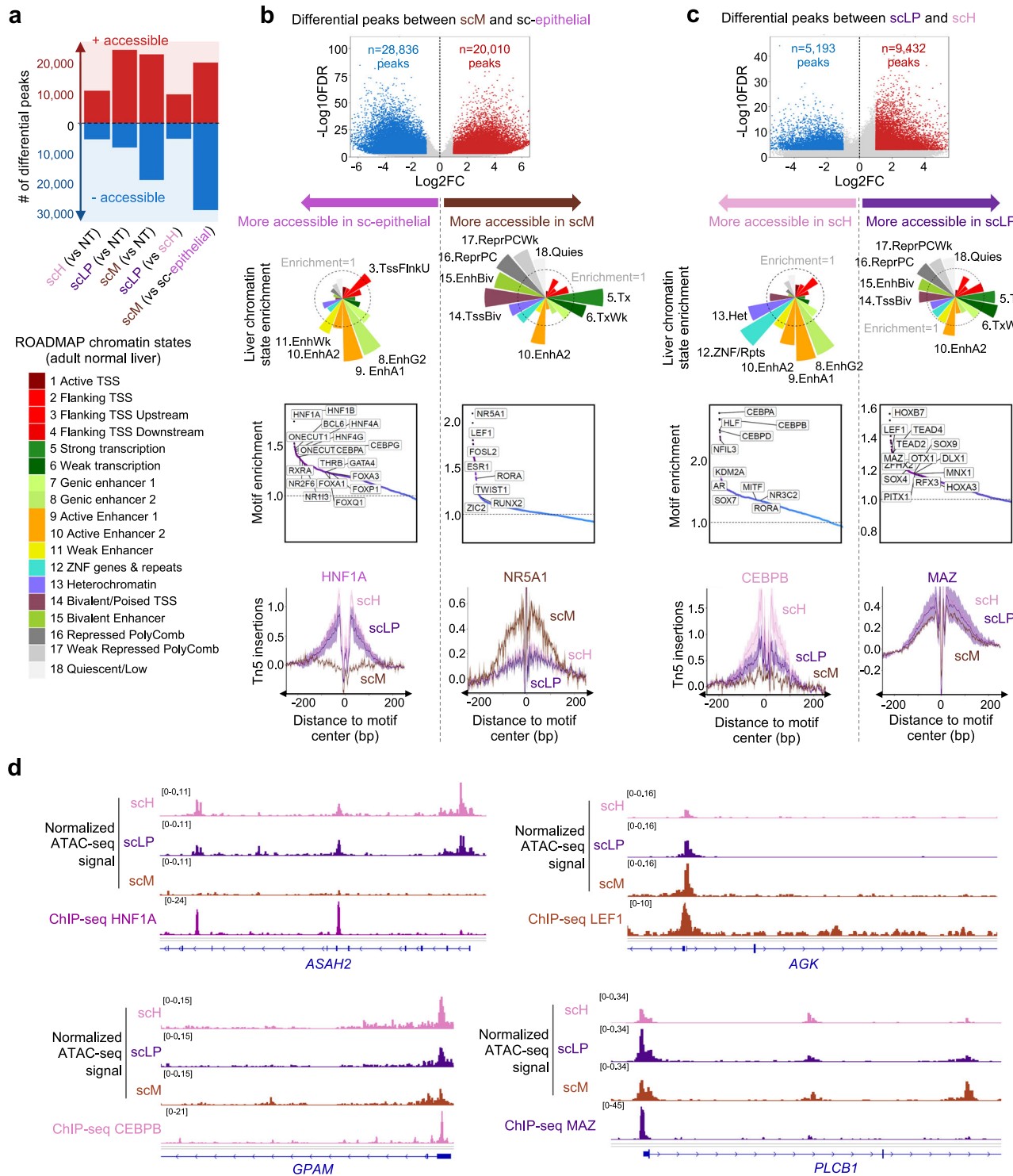

**Fig. 4 | Chromatin remodeling between single-cell states. a** Number of peaks with significantly different chromatin accessibility between scH (*n* = 2526), scH/LP (*n* = 4013), scLP (*n* = 2436), scM (*n* = 1964) tumor cells and non-tumor hepatocytes from the two non-tumor liver samples (NT, *n* = 1518). 'sc-epithelial' corresponds to the union of scH, scH/LP and scLP. Activated peaks are colored in red, repressed peaks in blue. **b** Characterization of differential peaks between scM and sc-epithelial cells. From top to bottom: Volcano plots showing the FDR-adjusted *p* value as a function of the log fold-change (*ArchR getMarkerFeatures* test), with significantly ($|\log_2FC| \geq 1$ & FDR $\leq 10^{-3}$) activated (resp. repressed) peaks show in red (resp. blue); Enrichment of normal liver chromatin states (from ROADMAP consortium) in differential peaks (the dotted circle represents enrichment=1); Transcription factor binding motifs enriched in differential peaks; Footprints of normalized Tn5 insertions around transcription factor binding motifs enriched in

sc-epithelial (HNF1A) and sc-M peaks (NR5A1). **c** Characterization of differential peaks between scLP and scH cells, formatted as in (**b**). **d** Significant overlap between differentially accessible ATAC-seq peaks and transcription factor ChIP-seq peaks from ReMAP database. The 4 examples shown include HNF1A associated with sc-epithelial cells (overlap between HNF1A ChIP-seq peaks and ATAC-seq peaks more accessible in sc-epithelial cells: *q* = 3.0e−122), LEF1 with scM cells (*q* = 1.9e−15), CEBPB with scH cells (*q* = 4.5e−6) and MAZ with scLP cells (*q* = 1.4e−5). Additional examples are shown in Supplementary Fig. 11, and all significant associations are reported in Supplementary Data 7. ChIP-seq coverage tracks were obtained from ENCODE (HNF1A in HepG2 cells: ENCFF502ACF; CEBPB in HepG2 cells: ENCFF406BBU; MAZ in HepG2 cells: ENCFF527EZL) or GEO (LEF1 in hESC: GSM1579343). Source data are provided in the Source Data file.

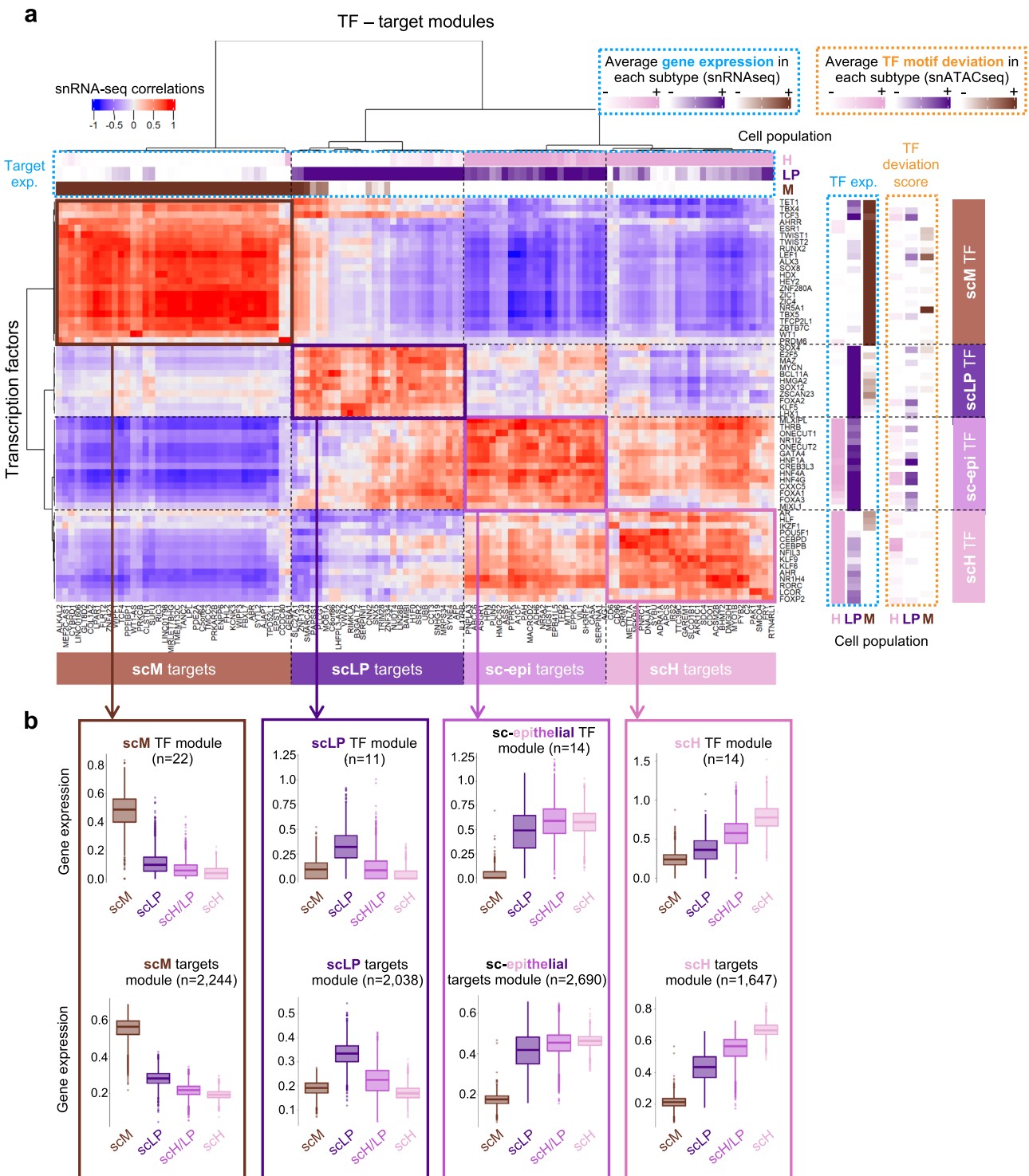

**Fig. 5 | Gene regulatory networks defining the 'scH', 'scLP' and 'scM' differentiation poles. a** Gene regulatory networks of HB differentiation cell states. The heatmap shows Pearson correlation coefficients between the expression of each transcription factor (TF, row) and target gene (column). Hierarchical clusterings revealed groups of co-regulated TFs and targets, as shown on the left and top dendograms. Labels above the heatmap indicate the mean expression (snRNA-seq) of target genes in each cell state. Labels on the right indicate the mean expression (snRNA-seq) of TFs and their motif deviation scores (snATAC-seq) in each cell state. TF-target modules were named according to the cell state in which they are more active. 'sc-epithelial' (or 'sc-epi') stands for the union of scH, scH/LP and scLP states. **b** Box-and-whisker plots showing the distribution of the mean expression of TFs (top) and target genes (bottom) of each module in each cell state. Middle bar, median; box, interquartile range; bars extend to 1.5 times the interquartile range. Source data are provided in the Source Data file.

mutations ($n = 26$) were restricted to specific clusters, matching the 2 samples of origin. Pseudo copy-number profiles also revealed trunk and subclonal CNAs specific to cl1 or cl2 (Fig. 8c). Integrating mutations and CNAs, we could thus reconstruct the evolutionary tree of this

patient and identify cells belonging to each subclone (Fig. 8d). We used the same strategy to reconstruct the single-cell clonal architecture of each HB and compare the phenotypic characteristics of genetic subclones (Fig. 8e, f and Supplementary Fig. 15).

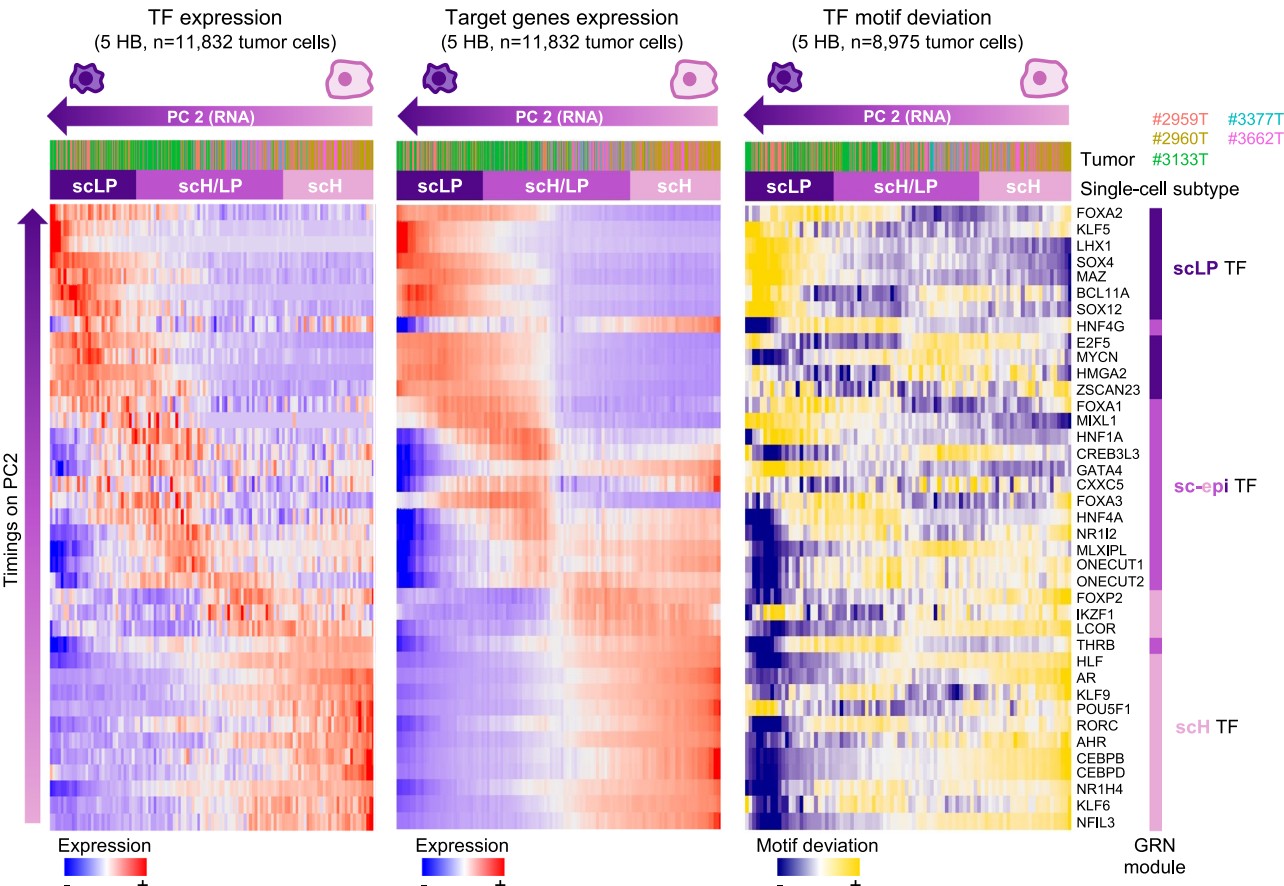

**Fig. 6 | Progressive activation of transcription factors along the scLP-scH differentiation axis.** Heatmaps showing the single-nucleus expression of transcription factors (TF, left), the average expression of their target genes (middle) and their motif deviations from snATAC-seq (right) along the scLP-scH axis. ScH, scH/LP and scLP cells (in columns) were ordered in each panel according to snRNA-seq PC2. TFs (in rows) were ordered according to their peak of activation along PC2. Cells are annotated by sample of origin and single-cell state. TFs are annotated according to the gene regulatory network they belong to (Fig. 5a).

## Interplay of clonal evolution and cellular plasticity

We first examined the differentiation states of cells within and between genetic subclones, as represented on individual UMAPs (Fig. 8d–f) and summarized in Fig. 8g. All subclones displayed some level of transcriptomic plasticity, but the breadth of differentiation states, and the orientation towards scH or scLP poles differed significantly between subclones. For example, patient #2959 cl1 displayed very diverse differentiation states, ranging from the most scH to the most scLP profiles, whereas cl2 displayed a more focused range with mostly scLP cells. In each tumor, the subclone with the most scLP differentiation had the highest proportion of cycling cells (Fig. 8g). A representative example is tumor #3662 T (patient #3660), in which we identified a small subclone, representing only 25 cells, with a high expression of scLP markers (Fig. 8f). This highly proliferative subclone was minor in the sample used for single-cell analysis (0.9% of cells), but it was dominant in the adjacent sample used for bulk WGS and RNA-seq (evidenced by clonal +2p, 8 and 12, Supplementary Fig. 5), which displayed a strong LP signature. Thus, single-cell clonal architectures can capture emerging subclones with a selective advantage at the time of sampling. In this series of surgical resections following neo-adjuvant chemotherapy, the most proliferative subclones are likely the most chemoresistant. Accordingly, scLP subclones displayed significant overexpression of DNA repair genes and cancer stem cell markers (Fig. 8g), two features associated with chemotherapy resistance[26,27]. Overall, each subclone displays its own range of phenotypic plasticity, likely driven by the genetic and epigenetic background of the common ancestor cell. The most LP-oriented subclones resist to chemotherapy and proliferate faster after neo-adjuvant treatment.

## Discussion

The histological diversity of HB was described a long time ago[28], but the molecular mechanisms underlying these heterogeneous cell states remained poorly understood. By analyzing the single-nucleus transcriptome and chromatin accessibility of selected HB representative of the main transcriptional and histological subtypes, we could define the key transcription factors and GRNs of HB cell states. We identified two main sources of variation in HB cell transcriptomes, corresponding to two differentiation axes between scM, scLP and scH poles. This analysis provides a more subtle description than our previous bulk RNA-seq study[5]. Interestingly, our deconvolution analysis shows that single cell states can be predicted from bulk data and may provide a more accurate description of HB transcriptomes, taking into account intra-sample heterogeneity. Spatial transcriptomics allows to visualize transcriptomic states together with histology. In tumor #3133 T, we found a good match between scH and fetal cells, and scLP and embryonal cells. Interestingly, HB cells with intermediate states were localized at the border between the two histological contingents. This analysis was however limited to a single case. Future studies of larger cohorts will be needed to explore the spatial organization of HB cell states and their interaction with their microenvironment with statistical rigor.

HB cell states are correlated with Cairo's signatures of early (scM) and late (scH) prenatal liver development[8] (Supplementary Fig. 16). More precisely, scM cells resemble the 'hepatomesenchymal' cell type identified by Lotto *et al.* at embryonic day E10.5[29]. Among epithelial HB cells, liver-specific cellular pathways are activated progressively between scLP and scH cells, mirroring their timing of activation in

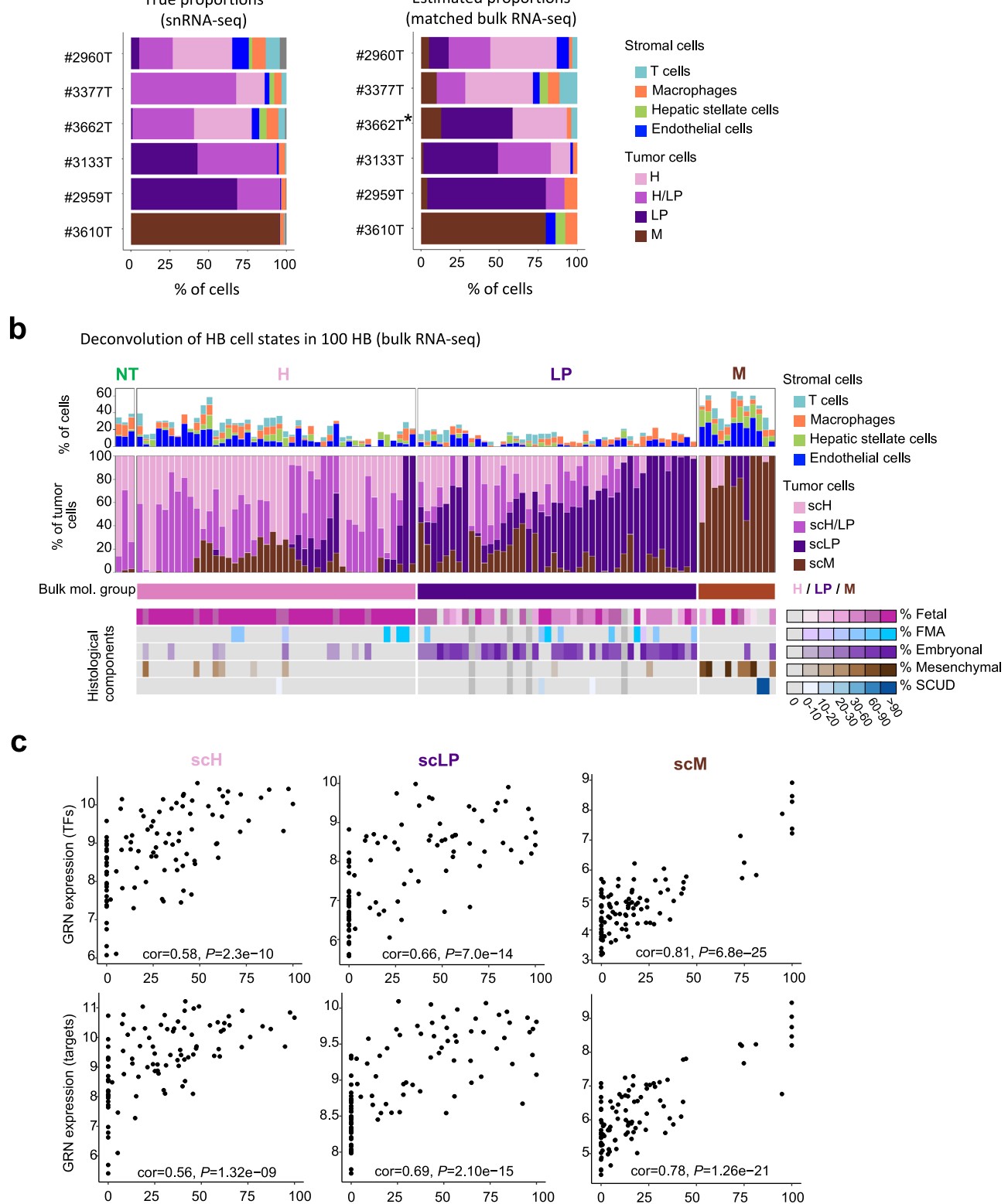

**Fig. 7 | Deconvolution of hepatoblastoma cell states in 100 bulk RNA-seq profiles. a** Comparison of cell proportions identified in single-nucleus data or our 6 HB samples (left) and those estimated by Bisque deconvolution tool in matched bulk RNA-seq profiles (right). **b** Deconvolution of HB cell states from bulk RNA-seq profiles of 100 HB samples (Hirsch data set[5]). Two barplots indicate the % of stromal cells (top) and tumor cell states (middle) in each sample. Bulk transcriptomic subgroups and histological components identified in mirror blocks are annotated below. **c** Correlation (Pearson's correlation test, two-sided) between the proportions of scH, scLP and scM cells estimated by Bisque and the expression of their gene regulatory networks (GRNs). Source data are provided in the Source Data file.

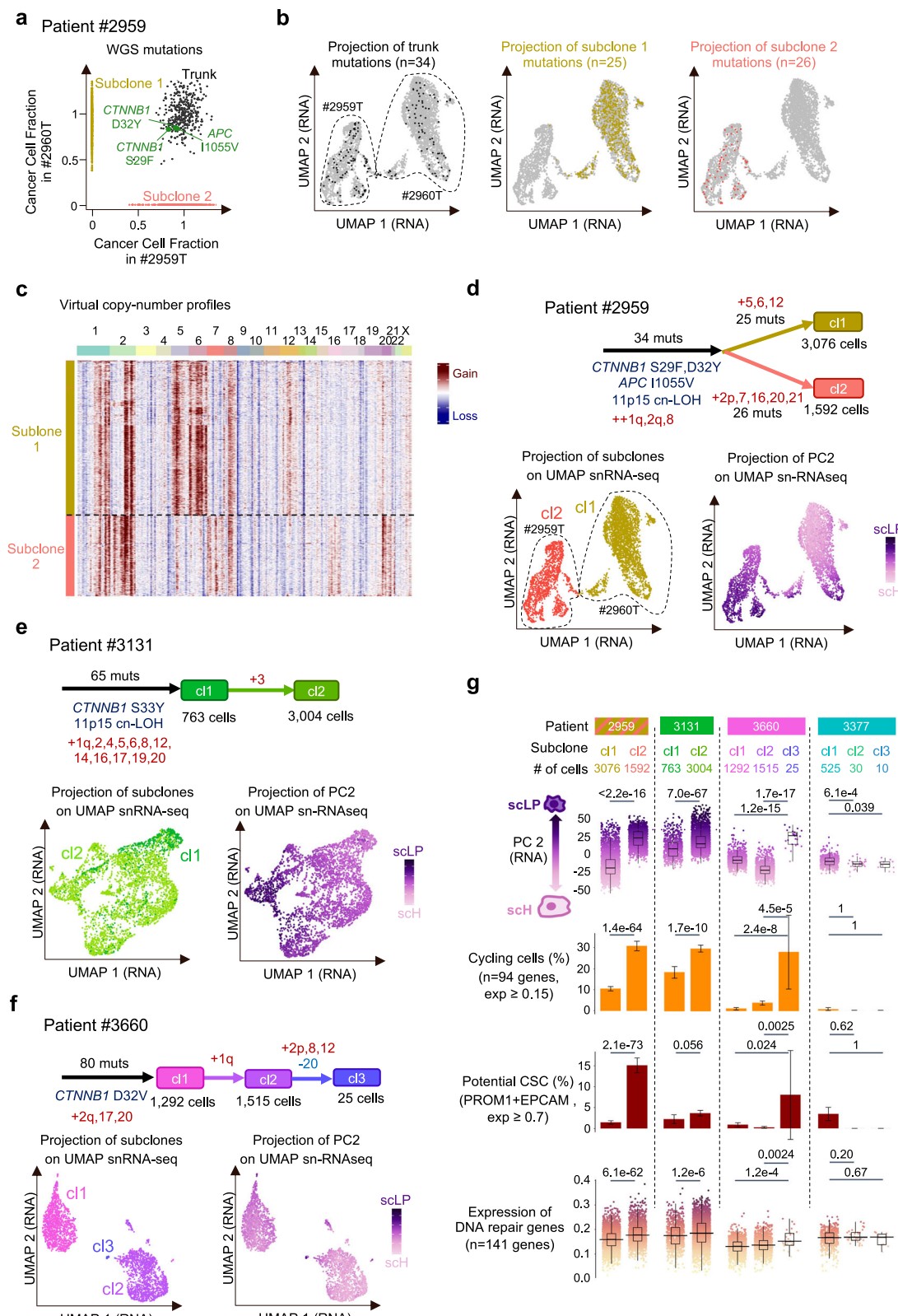

normal hepatocyte differentiation[30]. For example, genes characteristic of the latest differentiation stage ('adult hepatocyte' in Wesley et al. data), including xenobiotic metabolism and bile acid synthesis pathway genes, are mostly expressed in scH cells. Thus HB cell states reflect the different steps of normal liver development. They also show similarities with HB cell subtypes identified by Huang et al. (1293 tumor cells from 5 HB patients)[12] and HB tumor signatures identified by

Song et al. (6244 tumor cells from 9 HB patients)[13]. More precisely, our scM state is similar to Huang's HB2 (mesenchyme-like) group and expresses Song's 'fibroblast-like' signature. Our scLP subtype is closer to Huang's HB1 (progenitor-like) group, and our scH state has the highest expression of Huang's HB3 (hepatocyte-like) and Song's 'Hepatoblast' markers (Supplementary Fig. 16). However, by analyzing a larger number of cells (14,448 tumor cells in our study) from samples

**Fig. 8 | Interplay of clonal evolution and cellular plasticity in HB. a** Clonality of somatic mutations identified by whole genome sequencing in patient #2959. Each point represents a mutation, with its cancer cell fraction (CCF) in #2959 T on the x-axis, and in #2960 T on the y-axis. Trunk mutations (CCF ~ 1 in both samples) are highlighted, together with subclone 1 and 2 mutations, specific to sample #2960 T and #2959 T, respectively. **b** Projection of trunk (left), subclone 1 (midle) and subclone 2 (right) mutations on the snRNA-seq UMAP of patient #2959. The two main clusters in the snRNA-seq UMAP correspond to cells coming from each sample, as indicated with the dashed line. **c** Virtual copy-number profiles reveal 2 subclones matching mutation subclones. **d** Tumor progression tree reconstructed for patient #2959 by integrating WGS and single-nucleus data. Copy-number alterations and driver mutations are indicated on each branch, as well as the number of cells belonging to each subclone. Below, snRNA-seq UMAPs are annotated by genetic subclone (left) and PC2 contribution (right) indicating the level of scLP/scH differentiation. **e, f**, Same as (**d**) for patients #3131 and #3660. The tumor progression trees of the 2 remaining cases are shown in Supplementary Fig. 15, together with the projection of genetic subclones on snATAC-seq UMAPs. **g** Phenotypic characterization of genetic subclones (grouped by patient). From top to bottom: Distribution of PC2 contributions indicating the level of scLP/scH differentiation; Proportion of cycling cells based on the expression of G2/M and S phase marker genes; Proportion of potential cancer stem cells (CSC) based on the expression of PROM1 and EPCAM markers; Mean expression of 141 DNA repair genes ('Hallmark_DNA_repair' gene set from MSigDB[56]). Box-and-whisker plots: middle bar, median; box, interquartile range; bars extend to 1.5 times the interquartile range. Barplot error bars represent the 95% confidence intervals. *P* values were obtained using Wilcoxon rank sum tests (for PC2 and expression of DNA repair genes) or Fisher's exact test (for the proportion of cycling cells and CSC). All tests were two-sided. Source data are provided in the Source Data file.

representative of the main bulk molecular groups, and applying a dimension reduction technique less sensitive to tumor-specific effects, we unraveled a continuum of states between scLP, scH and scM cells. In particular, we defined an intermediate population of scH/LP cells expressing both H and LP markers at low levels, which we validated in spatial transcriptomics. ScH/LP cells do not correspond to a separate population, but they display a continuum of differentiation states between scLP and scH. Similar continuous profiles and intermediate states were previously observed in other cancers like glioblastoma, melanoma or colorectal cancer[31–33]. In this work, we focused on the plasticity between differentiation states, which were summarized by the first two principal components of snRNA-seq data. However, other principal components also account for a substantial part of the variance and certainly contain valuable biological information. Future analyses will be useful to extract the exhaustive catalog of transcriptional modules in HB.

Using snATAC-seq, we explored the epigenomic landscape of HB at single-cell level. The accessibility of peaks linked with liver differentiation genes increased from scLP to scH cells, indicating progressive engagement into hepatocytic differentiation. By contrast, scM cells displayed opening of bivalent chromatin domains opened in embryonic stem cells but that become repressed in differentiated liver. Motif analysis and co-expressed TF modules allowed us to reconstruct the GRNs associated with each cell state and their progressive activation during cell state switches. The intermediate population of scH/LP cells displayed a specific chromatin accessibility profile, with moderate accessibility of both scLP and scH markers. If this intermediate state was just a transient state of cells transitioning between the H and LP poles, we would expect a bimodal distribution along the PC2 axis, with only few intermediate cells. Rather, we observed in most samples a wide distribution of cells along PC2. This is in agreement with the idea that epigenetic remodeling mediates plasticity by reshaping gene regulatory networks into larger 'potential wells' or 'attractors'[34].

Single-cell sequencing data also give access to somatic alterations, including copy-number alterations and mutations. We identified at the single-cell level key genetic alterations in HB, like *CTNNB1* mutations and cn-LOH of the 11p15.5 locus. Leveraging matched WGS data, we could reconstruct the evolutionary tree of each tumor, and explore the interplay of clonal evolution and phenotypic plasticity. We found that transcriptomic cell states do not follow genetic subclones. Instead, each subclone remains plastic and displays a gradient of differentiation states. However, the extent of transcriptomic plasticity and the orientation towards more scLP or scH states is specific to each subclone. These data suggest that the genetic and epigenetic profiles of the last common ancestor cell determine the differentiation capacities of daughter cells within a given space. Given the variety of low frequency genetic and epigenetic driver alterations in HB[5,6], larger single-cell series will be needed to identify the molecular determinants driving HB subclones towards scLP differentiation. This is all the more important since scLP cells show the fittest phenotype after neo-

adjuvant chemotherapy, with a high proportion of cycling cells and expression of stem cell and DNA repair genes. Although tumor progression trees indicate a selection of the most 'scLP' subclones during neo-adjuvant chemotherapy, some metastases in our bulk RNA-seq cohort are mostly hepatocytic. Thus, HB cells could re-differentiate into 'scH' states after chemotherapy, when the pressure toward 'scLP' states is relieved. Longitudinal analyses will be useful to understand the cell state transitions occurring at each stage of the disease.

A limitation of this study is the small sample size. Single-cell multiomic analyses of larger series may reveal additional HB cell states not represented in the current data set. In addition, ATAC-seq only gives access to chromatin accessibility profiles. Profiling other epigenetic features, like DNA methylation[35] or histone modifications[36], will be useful to understand how the different layers of epigenetic regulation are orchestrated and their relative roles in HB cell plasticity.

## Methods

This research projects complies with all relevant ethical regulations. Written informed consent was obtained in accordance with French legislation, and the study was approved by the local Ethics Committee (CCPRB Paris Saint-Louis).

### Clinical samples

A series of 6 hepatoblastoma (HB) samples and 5 non-tumor liver counterparts were collected from 5 patients for this single-cell study. These samples were part of a larger cohort of 100 HB analyzed with bulk RNA-seq (tumor only) and whole genome or exome sequencing (matched tumor and non-tumor samples)[5], that was used for validation. All pediatric patients agreed to join the study and their parents / legal guardians provided signed informed consent (without compensation). Sex was not considered in the study design. The sex of participants was determined based on self-report. All samples were immediately frozen in liquid nitrogen and stored at −80 °C. Tumors were reviewed by three expert pathologists and fractions of histological components (fetal, embryonal, mesenchymal, cholangioblastic, and small cell undifferentiated) were estimated according to the consensus classification[7], for the whole tumor and for mirror blocks of frozen samples when available.

### Single-nucleus multiome

Single-nucleus Multiome (RNA + ATAC sequencing) was performed by Integragen SA (Evry, France) on 6 HB samples and 2 non-tumor liver counterparts from 5 patients. Single nucei were isolated using EZ Lysis buffer workflow with slight modifications[37], and nucleus permeabilization for ATAC-seq was performed following the application note of 10X Genomics (https://assets.ctfassets.net/an68im79xiti/4KtFk3LHb8UcgastMts0Mh/78449e967333569ce3989c93ad111ef7/CG000375_DemonstratedProtocol_NucleiIsolationComplexSample_ATAC_GEX_Sequencing_RevB.pdf). Briefly, tissue samples were thawed in PBS and cut into pieces <0.5 cm. Approximately 35 mg of tissue was

poured in a glass Dounce tissue grinder (Sigma, cat. no. D8938) and homogenized 25 times with pestle A and 25 times with pestle B in 1.5 ml of ice-cold nuclei EZ lysis buffer. Samples were then incubated on ice for 5 min with an additional 3 ml of cold EZ lysis buffer. Nuclei were centrifuged at 500 g for 5 min at 4 °C, washed with 5 mL ice-cold EZ lysis buffer, and incubated on ice for 5 min. After centrifugation, the nucleus pellet was washed with 1 mL of 10X-Genomics Wash buffer (containing 10 mM Tris-HCl pH 7.4, 10 mM NaCl, 3 mM MgCl2, 1% BSA, 1 mM DTT, 1 u/μL RNase Inhibitor (Sigma), 0.1% Tween 20) and filtered through a 70 μm then a 30μm MACS SmartStrainers (Miltenyibiotec 130-098-462 & 130-098-458). Then for permeabilization, the pellet was resuspended in 100 μL of 0.1X Lysis buffer (10 mM Tris-HCl pH 7.4, 10 mM NaCl, 3 mM MgCl2, 1% BSA, 1 mM DTT, 1 u/μL RNase Inhibitor (Sigma), 0.01% Tween 20, 0.01% Nonidet P40, Digitonin 0.001%), and incubated on ice for 2 min. After incubation, 1 ml of 10X-Genomics wash buffer was added and permeabilized nuclei were centrifuged at 500 g and washed twice again in 10X Genomics wash buffer. Finally after centrifugation the pellet was resuspended in 300 μl of 10X Genomics diluted Nuclei Buffer (1X nuclei buffer, 1 mM DTT, 1 u/μL RNase Inhibitor (Sigma)) and nuclei were counted under microscope using C-chip disposable hemocytometer. A final concentration of 1000 nuclei per μL was used for the Chromium Next GEM Single Cell Multiome ATAC + Gene Expression kit, following the CG000338 user guide for transposition, GEM generation, ATAC and Gene Expresssion library construction. 7700 nuclei were loaded on Chromium to target 5000 recovered nuclei.

## Multiome alignment

We used cellranger-arc (v2.0.0)[38,39] to align snRNA-seq and snATAC-seq reads to the human genome (GrCh38/hg38) and generate the matrices of UMI counts per gene and ATAC-seq fragments. We modified the genes.gtf file used for snRNA-seq analysis by removing genes without HGNC ID that overlap HGNC genes. This prevents cellranger-arc from discarding reads overlapping well established protein-coding genes and non-coding RNAs (e.g. antisense), as cellranger-arc discards reads mapping to several genes in the gtf file. Our modified gtf included 35,010 genes (vs. 36,601 for the default gtf file). After alignment, we obtained a total of 21,921 cells (median 2489 per sample, see Supplementary Data 2 for detailed metrics).

## snRNA-seq pre-processing and visualization

We filtered the feature-barcode gene expression matrix to keep only high quality nuclei (>1000 UMI counts, >500 detected genes and <5% of mitochondrial reads) and expressed genes (detected in ≥3 nuclei). After QC, we obtained a total of 21,150 cells (median 2348 cells per sample with 8408 UMIs and 3263 genes per cell, see Supplementary Data 2 for extensive metrics). Secondary analyses were performed using Seurat (v3)[40]. For each sample, we normalized the filtered UMI count matrix with SCTransform (default parameters). We used the 3000 most variable genes to perform principal component analysis with runPCA, and ran Louvain graph-based clustering on the 30 first principal components (using FindNeighbors and FindClusters with a resolution of 0.3) before projecting the resulting data on a 2D UMAP with runUMAP. The same strategy was applied to the merged dataset containing all nuclei from the 6 HB and 2 non-tumor liver samples ($n = 21,150$), with 40 principal components for clustering and UMAP visualization. Finally, we log-normalized the raw counts with Seurat NormalizeData (for each cell, UMI counts in each gene are divided by the total cell counts and multiply by 10,000 before being natural log-transformed) for gene expression quantification.

## snATAC-seq pre-processing and visualization

snATAC-seq data analysis was conducted with the ArchR (v1.0.1)[41] package. We first examined the distributions of number of fragments per nucleus and transcription start site (TSS) enrichment in each sample, and we applied ad hoc filters (see Supplementary Data 2) to retain only the most reliable cells for further analysis. After QC, we obtained a total of 17,649 cells (median of 2071 cells per sample, with 10,027 fragments per cell and a TSS enrichment of 9.68, see Supplementary Data 2). We used ArchR createArrowFiles function with GrCh38/hg38 genome reference to generate sample Arrow files, the tile matrix containing insertion counts every 500-bp genomic tile, and the gene score matrix containing expression predictions based on weighting insertion counts in tiles nearby gene promoters. We used addDoubletScores to identify potential doublets resulting from the encapsulation of two nuclei in the same droplet. To visualize the proximity of single-nucleus chromatin accessibility profiles, we applied the addIterativeLSI function ro perform TF-IDF normalization on the tile matrix, and we projected the resulting data on a UMAP with addUMAP (using 30 nearest neighbors). The same strategy was applied to individual samples and to the merged dataset containing all nuclei from the 8 samples.

## snATAC-seq peak calling

We used Signac[42] v1.6 CallPeaks function (which relies on the MACS2 peak caller[43]) to identify chromatin accessibility peaks in each sample. Peak calling was performed by cell state (scH, scLP, scH/LP and scM) to increase state-specific peak detection. We removed blacklisted peaks (manually curated from ENCODE) and those on sex chromosomes, and we combined the 6 resulting peak sets with the reduce function (GenomicRanges package[44]) that merges intersecting peaks. Removal of outlier peaks (smaller than 20 bp or larger than 10,000 bp) resulted in a final set of 158,707 peaks (median width 451 bp, range 200-3824 bp). Finally, we used ArchR addPeakMatrix to generate the peak-cell count matrix.

## Virtual copy-number profiles

We reconstructed virtual copy-number profiles from the raw snRNA-seq counts of the 21,150 nuclei to distinguish tumor from normal nuclei, and identify tumor subclones with specific copy-number changes. To that aim, we used InferCNV[45] package (version 1.6) with default parameters, keeping genes with an average read count > 0.1 in reference nuclei. We used as reference nuclei healthy hepatocytes from the non-tumor sample #2959 N (n = 719 nuclei) and inferred the virtual CNV profiles for the remaining 20,431 nuclei. InferCNV clustering was used to identify non-tumor cells and tumor cell clusters with similar copy-number profiles.

## Cell type identification

We used transcriptomic markers of HB cells[5] and microenvironment populations to annotate Seurat clusters. We also used InferCNV clusters to identify cells with copy-number alterations. We annotated as tumor cells those belonging to Seurat clusters expressing HB markers, and to InferCNV clusters with copy-number alterations ($n = 14,448$). Microenvironment cells were annotated based on the expression of the following markers: CD247 for T cells, CDH5 for endothelial cells, COL3A1 for hepatic stellate cells and CD163 for macrophages. Remaining undefined cells (~6% of all nuclei) belonged to clusters with high doublet enrichment scores, showing both expression of immune markers and abnormal virtual copy-number profiles. These clusters likely correspond to doublets and were removed from the analysis.

## Batch effect estimation

Principal component (PC) regression analysis was previously used to estimate the proportion of variance in a single-cell data set explained by sample-specific variations, which may indicate the presence of technical batch effects[46]. Distinguishing technical from biological sample-specific variations is challenging in cancer since tumor cells from different patients display specific driver alterations and copy-number changes. Non-tumor cells are expected to be more similar

across samples, but it is important to keep in mind that tumors also differ by the abundance and composition of their immune infiltrates. In HB, we previously showed that APC-mutated HB have massive intratumor tertiary lymphoid structures[47] and that liver progenitor HB samples are "immune-cold" compared to hepatocytic HB samples[5]. We thus expect that some of the patient-related variance will be of biological origin, even in non-tumor cells. However, we used PC regression to quantify the proportion of variance explained by sample of origin in our data set. We focused on the first 5 principal components (PC1-5) that together account for 61.5% of the variance in non-tumor cells. We first used linear regression (lm function in R statistical software) to estimate the proportion of each PC explained by cell type: R2(PCi,cell type). We then estimated the proportion of PC1-5 variance explained by cell type as the sum of R2 values, weighted by the variance of each PC, divided by the total variance of PC1-5 (Eq. (1)).

$$\frac{\sum_{i=1}^{5} R2(PСi, cell\ type) * var(PCi)}{\sum_{i=1}^{5} var(PCi)} = 0.686 \quad (1)$$

Thus, cell type explains 68.6% of the variance of the first 5 PCs.

Next, we repeated the regression analysis including both cell type and sample of origin as explanatory variables. We estimated the proportion of each PC explained by both factors: R2(PCi,cell type & sample of origin). Finally, we calculated the proportion of PC1-5 variance explained by cell type and sample of origin (Eq. (2)).

$$\frac{\sum_{i=1}^{5} R2(PCi, cell\ type\ \&\ sample\ of\ origin) * var(PCi)}{\sum_{i=1}^{5} var(PCi)} = 0.828 \quad (2)$$

Thus, cell type and sample of origin explain 82.8% of the variance of the first 5 PCs. In other words, sample of origin explains 14.1% of the variance in addition to the 68.6% explained by cell type. This extra 14.1% of explained variance may be due to technical reasons (batch effect) and/or biological differences between the immune infiltrates of the samples.

## Classification of tumor nuclei

Using scPower (v1.0.4)[48], we estimated that, with the number of samples, cells per sample and reads per cell of our study, we have 95% power to identify a population representing 2.7% of all cells. Tumor nuclei ($n = 14{,}448$) were classified by UMAP, or simple projection over the 2 first components of the PCA. We visualized on these graphs the mean log-normalized expression of bulk 'H', 'LP' and 'M' markers[5], defined as the 100 genes with highest log fold-change in the respective group vs. other HB. PC1 was strongly correlated with the expression of 'M' markers, (cor=0.97, P < 2.2e−16), and PC2 was correlated positively with 'LP' markers (cor = 0.82, P < 2.2E−16) and negatively with 'H' markers (cor = -0.88, P < 2.2e−16). We then defined four single-cell HB subtypes based on the two first PCA components. Tumor cells with high PC1 contribution (corresponding to cells of the mesenchymal tumor #3610 T) were defined as the 'scM' subtype ($n = 2616$). The remaining epithelial tumor cells, scattered along the PC2 axis, were divided in 3 subtypes based on PC2 thresholds: scH (PC2 < -15, $n = 3281$ cells), scLP (PC2 > 15, $n = 3153$ cells) and the intermediate scH/LP subtype (-15 ≤ PC2 ≤ 15, $n = 5398$). With these cell numbers, we estimated with scPower that we have a detection power of differentially expressed genes of 0.63 for scH, 0.62 for scLP and 0.57 for scM cells. We used Seurat marker genes of G2/M ($n = 52$) and S phases ($n = 42$) to identify cycling cells (mean normalized expression of the 94 genes ≥ 0.15).

## Spatial transcriptomics

Spatial transcriptomics was performed according to 10X genomics Visium protocol with a 5 µM FFPE section of tumor #3133 T. Visium spots were manually annotated by an expert pathologist as tumoral, stromal or mixed based on the examination of H&E images in 10X Genomics Loupe Browser tool. Spots containing a mixture of tumor and stromal cells or hematopoiesis were excluded from the analysis. Two tumor nodules were identified and annotated as fetal or embryonal histology, and the spots at the interface between the two components were labeled as such. We used 10X Genomics spaceranger (v1.3.1) tool (mkfastq and count commands) to process raw sequencing data. The count matrix was analyzed with Seurat (v.4.2.0) package and normalized using SCTransform function. We calculated the average normalized expression of scH and scLP markers in each Visium spot. ScH (resp. scLP) markers were defined as the 120 genes with the highest positive (resp. negative) contributions to PC2 in the snRNA-seq data PCA (Supplementary Data 3). Of these, 91 scH and 102 scLP markers were detected in Visium data and used for this analysis. We used Seurat SpatialFeaturePlot function to project the expression of scH and scLP signatures on HE-stained digital slides, and to compare molecular data with histological annotations.

## Detection of somatic mutations in single-nucleus data

We used screadcounts (version 1.1.8)[49] to detect in snRNA-seq data the somatic mutations identified by WGS. Screadcounts returns for each mutation the number of altered (ALT) and reference (REF) reads in each cell. We used it to detect specific mutations, e.g. CTNNB1 driver mutations, or sets of mutations belonging to specific branches of the phylogenetic trees. Screadcounts was applied to snRNA-seq BAM files restricted to reads belonging to proper nuclei identified by cellranger-arc (options −G STARsolo −b cell IDs). To reduce false positives when analyzing sets of mutations, we further filtered the bam files with samtools view (version 1.14)[50] to retain only high-quality reads kept for UMI counting (cellranger-arc tag xf = 25). Nuclei were considered mutated for a set of mutations if they displayed at least 1 altered read for one of the mutations. For *CTNNB1* mutations, we removed from each sample the mutations that were detected in <3 nuclei.

## Detection of 11p15 copy-neutral LOH in single-nucleus data

We used germline single-nucleotide polymorphisms (SNPs) to detect in snRNA-seq data the copy-neutral LOH of 11p15 locus identified by WGS. For each sample, we used WGS data to delimitate the LOH, and assign the paternal (PAT, duplicated) and maternal (MAT, lost) allele of each heterozygous SNP in the region. The status was assigned based on the B Allele Frequency (BAF): PAT = B if BAF > 0.65; PAT = A if BAF < 0.35. We then used screadcounts (version 1.1.8)[49] to count the number of PAT and MAT alleles for each SNP in each cell. Screadcounts was applied to snRNA-seq BAM files restricted to reads belonging to proper nuclei identified by cellranger-arc (options −G STARsolo −b cell IDs). We summed the number of PAT and MAT reads in each cell, and we used a binomial test (with greater alternative) to estimate the probability of observing these numbers in absence of LOH (expected proportion of PAT reads = 0.5). Nuclei with a probability <0.05 were considered to harbor the copy-neutral LOH. Others were assigned a normal 11p15 status if the total count of MAT reads was ≥3, and unknown otherwise. Finally, we estimated a pseudo-BAF at the sample level as the sum of PAT alleles divided by the total number of PAT and MAT alleles over all cells.

## Chromatin accessibility landscape of HB differentiation poles

We used ArchR getMarkerFeatures function (maxCells=1000 nuclei per group) to identify differentially accessible peaks between differentiation poles. ScH ($n = 2526$ nuclei with post-QC ATAC-seq profiles), scLP ($n = 2436$) and scM ($n = 1964$) nuclei were compared between them and against hepatocytes from the 2 non-tumor samples #2959 N and #3377 N ($n = 1518$). Significant differential peaks ( |log₂FC| ≥ 1 & FDR ≤ 10⁻³) are recapitulated in Supplementary Data 5. Differential peaks were projected on volcano plots using the markerPlot function.

We examined the enrichment of differential peaks over the 18 chromatin states defined by the ROADMAP consortium in normal adult liver[18]. For each chromatin state, the enrichment was calculated as the ratio between the proportion of nucleotides in that state in differential peaks, and the same proportion in all peaks.

### Transcription factor binding motifs

We used ArchR functions to compute TF motif deviations indicating nuclei with high/low accessibility of the motif. We first used addImputeWeights to impute weights on snATAC-seq data, and addBgdPeaks (parameter method = "ArchR") to identify background peaks with similar GC content and number of fragments across all samples. We then used addDeviationsMatrix to compute TF motif deviations across all nuclei ($n = 17,649$).

To identify TF motifs enriched in differential peaks, we calculated the ratio between the proportion of nucleotides intersecting the motif in differential peaks, and the same proportion in all peaks. We used two versions of cisbp database (http://cisbp.ccbr.utoronto.ca/)[51] to define the motifs: version 1 (870 motifs) and version 2 (1,065 motifs). Some motifs were modified between the two versions due to the addition of experiments. In that case, we kept the motif version with the highest enrichment score.

We performed TF footprinting to compare the accessibility of key TFs between HB cell states. We used ArchR addGroupCoverages function to aggregate snATAC-seq profiles of scH, scLP and scM cells into pseudo-bulk samples. We then used getFootprints to extract TF footprints, and plotFootprints (parameters: flank=250, flankNorm=50, baseSize=10, smoothWindow=5) to visualize the Tn5 bias-substracted TSS insertion profiles.

### Computation of aggregated single-nucleus expression data

Single-cell data is by essence sparse due to the abundance of zeros. For some analyses, we aggregated the 14,448 tumor nuclei into 145 small clusters with similar transcriptomes using Seurat FindNeighbors on 40 dimensions followed by FindClusters with Resolution = 20. Molecular features (log-normalized gene expression, PC1 or PC2 values, TF motif deviations…) were averaged within these clusters to obtain smoother, more interpretable results, not blurred by a majority of zeros.

### Identification of key TFs defining HB differentiation states

To identify key regulators of HB cell states, we computed composite scores integrating information from this snMultiome series and bulk RNA-seq of 100 HB[5]. The goal was to leverage single-cell resolution data of TF expression and motif accessibility, while ensuring that the findings were validated in a large HB series. Of the 1639 transcription factors (TFs) defined by Lambert et al.[52], 1514 were expressed in our snRNA-seq data. We computed for each TF the following metrics:

(1) Bulk RNA-seq differential expression (log fold-change) between 'H', 'LP' and 'M' groups
(2) Correlation of snRNA-seq expression with PC1 and PC2
(3) Correlation of snATAC-seq motif deviation with PC1 and PC2.

In practice, we used bulk RNA-seq 'M' vs 'H and LP' comparison, and single-nucleus correlations with PC1, to define TFs regulating scM and sc-epithelial (union of scH, schLP and ssH/LP cells) cells. We used bulk RNA-seq 'LP' vs 'H' comparison, and single-nucleus correlations with PC2, to define TFs regulating scLP and scH cells. Log-normalized TF expression and TF motif deviations were aggregated by small cell groups to avoid missing data, as explained in the 'Computation of aggregated single-nucleus expression data' section.

Finally, we calculated the percentile of each significant TF (q-value ≤ 0.05) for each metric, and we obtained the composite score as the mean of the 3 percentiles.

### Gene regulatory networks and their timing

We selected the 20 TFs with the highest composite scores in scH, scLP, scM and sc-epithelial cells to reconstruct the gene regulatory networks (GRN) of HB, e.g. modules of co-activated TFs and their target genes. TFs without motif binding information (e.g. not represented in cisbp) were discarded, and we added 3 TFs previously identified in bulk RNA-seq as key regulators of the LP (MYCN, MIXL1) and M (TBX5) subtypes[5]. Overall, 61 key TFs were included (∼15 for each cell state). We then identified the target genes of each TF meeting the two following criteria:

1. Genes whose aggregated single-nucleus log-normalized expression was significantly correlated with that of the TF (correlation > 0.5, q-value ≤ 0.001).
2. Genes linked with an ATAC-seq peak comprising the TF binding motif. We used ArchR addPeak2GeneLinks to link genes with snATAC-seq peaks based on the correlation between gene expression and peak accessibility, and getPeak2GeneLinks (with parameters corCutOff = 0.4, FDRCutOff = 0.01, varCutOffATAC= varCutOffRNA = 0.25 & Resolution=1) to extract significant links. We obtained links between 39,409 peaks and 11,173 genes. We considered a gene as a potential target if it was linked to at least 1 peak containing the TF binding motif.

We identified 6663 target genes, with a median of 424 targets per TF (Supplementary Data 7 and 8). To define core TF-target modules, we then selected the top 2 targets showing the highest correlation with the TF in snRNA-seq data. We then used ComplexHeatmap[53] to perform hierarchical clusterings (Pearson distance, Ward.D method) of both TFs and their top target genes based on their correlation matrix. The clustering of the TFs revealed 4 modules (Fig. 5a), which we named scH, scLP, scM and sc-epi according to their level of activation in each cell state. The sc-epi and scH modules were quite close in terms of target genes, but they displayed different dynamics of activation between scLP and scH states (Fig. 5b) so we decided to keep them separated.

### Validation of key transcription factors using public ChIP-seq data

We used ChIP-seq data from the ReMap database[20] (https://remap2022.univ-amu.fr) to provide additional evidence supporting the master TFs of each cell state. The 4th release of ReMap (remap2022_nr_macs2_hg38_v1_0 version) compiles 8103 quality controlled ChIP-seq datasets from various public sources. We used the ReMapEnrich R package (v0.99.0) to systematically analyze the overlap between ATAC-seq peaks significantly more accessible in each cell state ($\log_2 FC > 1$ & FDR < $10^{-3}$) and TF ChIP-seq peaks from ReMap.

### Deconvolution of HB cell states in bulk RNA-seq data

We used Bisque deconvolution tool[25] to deconvolute HB cell states in bulk RNA-seq data from 5 published series, totaling 314 HB samples[5,6,10,11,17]. We used as reference a Seurat object containing the merge of the 8 snRNA-seq samples, restricted to the major cell types found in tumors (Endothelial, Liver stellate, Macrophages, T cells, Tumor scH, Tumor scH/LP, Tumor scLP and Tumor scM cells). This reference was fed to the deconvolute_all function from the deconverse R package (v0.2.0, https://github.com/csgroen/deconverse) with the Bisque method, applied to the expression matrix of bulk RNA-seq data normalized by variant stabilization (cohorts Hirsch 2021, Hooks 2018, Carrillo-Reixach 2020, Sekiguchi 2020) or FPKM (cohort Nagae 2021). We correlated the estimated proportions of scH, scLP and scM cells with the mean expression of their GRN TFs. We also used Wilcoxon rank sum tests to compare the abundance of tumor and stromal cell types between pre- and post-chemotherapy samples in the two cohorts that contained both (Sekiguchi 2020 and Hirsch 2021).

**Deconvolution of HB cell states in spatial transcriptomic data**

We used SPOTlight deconvolution tool (v1.5.1)[54] to deconvolute each spot of the Visium spatial transcriptomics data as a mixture of HB cell states. Briefly, we used as reference a Seurat object containing the merge of the 6 snRNA-seq tumor samples, restricted to the epithelial tumor cell states (scH, scH/LP and scLP), using down-sampling to keep only 500 cells of each type, as recommended in the SPOTlight manual. This reference was fed to the SPOTlight function, applied to the Seurat object of the Visium data restricted to the 2 tumor nodules of interest (regions 1 and 2 annotated in Fig. 3).

**Reporting summary**

Further information on research design is available in the Nature Portfolio Reporting Summary linked to this article.

## Data availability

Raw sequencing data from the single-cell Multiome and spatial transcriptomics experiments performed for this study have been deposited to the European Genome Archive (EGA) under accession code EGAS00001006932. These data contain identifiable genetic variants and are thus accessible under controlled access for patient privacy concerns, by contacting the data access committee. We also re-analyzed single-cell RNA-seq data from Song et al. (GEO database accession GSE186975) and bulk RNA-seq from Hirsch et al. (EGA accession EGAS00001005108), Nagae et al., (NBDC Human Database accession hum0233-v1), Hooks et al. (GEO database accession GSE104766), Sekiguchi et al. (Japanese Genotype-phenotype Archive accession JGAS000088R) and Carrillo-Reixach et al. (GEO accession GSE132219). ChIP-seq bigwig files were downloaded from ENCODE accession numbers ENCFF502ACF, ENCFF406BBU ENCFF527EZL, and GEO accession GSM1579343. Source data of Figures and Supplementary Figs. have been provided as Source Data files. Source data are provided with this paper.

## Code availability

Custom scripts generated for this study are available on Github (https://github.com/FunGeST/Roehrig2023_HB_plasticity_scripts) and have been Custom codes developed for this project have been deposited on Zenodo (https://doi.org/10.5281/zenodo.10610870) under GNU General Public License[55].

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

## Acknowledgements

We warmly thank members of the thesis committee of Amélie Roehrig (Josh Waterfall, Céline Vallot and Andrei Zinovyev) for fruitful discussions and advice. This work was supported by France Génomique (GEPELIN project), Institut National du Cancer (INCa, PELICAN.resist and PEDIAHR22-009 projects), SFCE, association Etoile de Martin, Fédération Enfants et Santé (FES), association Hubert Gouin « Enfance et Cancer », INSERM with the HTE (plan Cancer), Cancéropôle Ile-de-France « ThéBioCaPe » project, Inserm ITMO Cancer 3 R "PEILICANS" project. The group is supported by the Ligue Nationale contre le Cancer (Equipe Labellisée), Labex OncoImmunology (investissement d'avenir, France 2030), grant IREB, Coup d'Elan de la Fondation Bettencourt-Shueller, the SIRIC CARPEM, FRM prix Rosen, Ligue Contre le Cancer Comité de Paris (prix René et Andrée Duquesne), Télévie, Fondation Nuovo Soldati, Fondation Mérieux, CisMutHep InCa High-Risk High-Gain (Institut National du Cancer, grant number PEDIAHR22-009) and THRIVE EU Horizon funding Mission Cancer (EU-HORIZON-MISS-2023-CANCER-01). AR was funded by Fondation pour la Recherche Médicale (FRM, grant number ECO201906008977), and Fondation ARC pour la recherche sur le cancer (ARCDOC42022010004699). TZH was funded by SIRIC CARPEM.

## Author contributions

A.R.: conceptualization, investigation, visualization, writing-original draft. T.Z.H.: investigation, visualization. A.P.: investigation. G.M.: investigation, visualization, resources. B.G.: investigation, visualization. C.M.: investigation. S.I.: investigation. C.C.: resources. E.G.: resources. E.J.: resources. M.S.: resources. J.T.: resources. G.N.: resources. E.H.: resources. F.G.: resources. M.F.: resources. I.A.: resources. S.T.: resources. V.L.: resources. S.B.: resources. C.G.: resources. L.B.: resources. B.F.: resources. J.Z.R.: conceptualization, supervision, investigation, funding acquisition, writing-original draft. E.L.: conceptualization, supervision, investigation, funding acquisition, writing-original draft.

## Competing interests

The authors declare no competing interests.

## Additional information

[1]Centre de Recherche des Cordeliers, Université Paris Cité, Sorbonne Université, INSERM, Paris, France. [2]Department of Pathology, Robert Debré and Necker-Enfants Malades Hospitals, APHP, Paris, France. [3]IntegraGen SA, Evry, France. [4]Université de Paris, APHP, Necker Hospital, Paris, France. [5]Pediatric Hepatology and Liver Transplantation Unit, National Reference Centre for Rare Pediatric Liver Diseases, FILFOIE, ERN RARE LIVER, APHP, Bicêtre University Hospital, University of Paris-Saclay, Le Kremlin Bicêtre, and INSERM UMR_S 1193, Hepatinov, University of Paris-Saclay, Orsay, France. [6]Department of Pediatrics, Graduate School of Medicine, The University of Tokyo, Tokyo, Japan. [7]Department of Pediatrics, Graduate School of Medicine, Kyoto University, Kyoto, Japan. [8]Genome Science Laboratory, Research Center for Advanced Science and Technology (RCAST), the University of Tokyo, Tokyo, Japan. [9]Department of Pediatric Surgery, Hiroshima University Hospital, Hiroshima, Japan. [10]Department of Biomedical Science, Natural Science Center for Basic Research and Development, Hiroshima University, Hiroshima, Japan. [11]Department of Pediatric Surgery, Bicêtre Hospital, APHP, Paris-Saclay University, Orsay, France. [12]Department of Pathology, Hôpital Universitaire Necker-Enfants malades, AP-HP, Paris, France. [13]Oncology Center SIREDO, Institut Curie, PSL Research University, Paris, France. [14]Département de Pédiatrie, CHU Fontenoy, Rennes, France. [15]Department of Children Oncology, Centre Hospitalier Universitaire Besançon, Besançon, France. [16]Department of Pathology Hôpital Bicêtre-AP-HP, INSERM U1193, Paris-Saclay University, Orsay, France. [17]Gustave Roussy, Université Paris-Saclay, Department of Children and Adolescents Oncology, Villejuif, France. [18]Hôpital Européen Georges Pompidou, Assistance Publique Hôpitaux de Paris, Paris, France. [19]CRCI2NA, Nantes Université, INSERM, CNRS, Nantes, France. [20]University Hospital Hôtel-Dieu, Nantes, France. [21]These authors contributed equally: Jessica Zucman-Rossi, Eric Letouzé. ✉e-mail: jessica.zucman-rossi@inserm.fr; eric.letouze@inserm.fr

