## [Peer review file · Nature Communications]

REVIEWER COMMENTS

Reviewer #1 (Remarks to the Author): Expert in hepatoblastoma clinical research and genomics

This paper presented the interplay of clonal evolution and epigenetic plasticity shapes and suspected the potential of HB subclones to respond to chemotherapy using mainly single cell sequencing of HB tissues. The data are very interesting and integrated but the data seems as the make-up examination of the previous report (ref. 5) by the authors (Hirsh T.Z. et al. Cancer Discov. 2021). If authors want to submit this paper, the difference between this paper and the previous one by Hirsh T.Z et al. should be clarified and the followings should be solved or reconsidered. This previous paper already reported the scSeq and the classification of scH, scLP, scM, and scH/LP.

If the authors want to show “the clonal evolution and cellular plasticity in hepatoblastoma tissues”. The difference of scSEQ and histological examination. In this paper, the results of single-cell multiomics only show the same finding of histology.

1. As mentioned in their previous paper, ‘Liver Progenitor’ cells accumulated massive loads of cisplatin-induced mutations with a specific mutational signature, leading to the development of heavily mutated relapses and metastases. Therefore, genomic alteration will be induced by chemotherapy such as CDDP and Doxorubicin. The authors should clarify the reason why these 6 samples obtained after preoperative CTX were used.
2. Moreover, why were these 6 cases selected from 100 cases including 18 tumors obtained at diagnosis? The authors should select the cases whose tumors were resected before chemotherapy.
3. Moreover, why did the authors examine only 2 non-tumorous liver tissues. The LOH analysis using WGS might require the individual normal tissue. Hirsch cohort had 18 normal tissues. The authors should add more data of non-tumorous liver tissues.

In addition

4. How did you detect “cnLOH” without the data of normal tissue?
5. In Figure 3, there are fibrosis and necrotic findings. How did you confirm the viable tumor cells in this specimen? It is possible that the scSeq showed only viability of the tumors
6. In ATAC seq data, the authors should describe its correlation with methylation status because the methylation of imprinting genes are well-known correlated to hepatoblastoma development.
7. In Fig 4a, what is “N”. Non-tumor samples were only 2. If you said that “N” is non-tumor hepatocytes in the tumor tissue, you have to decline how to detect them.

8. Each figures are too small to be understood. The authors should reproduce them to be understood easily.

Reviewer #2 (Remarks to the Author): Expert in hepatoblastoma clinical research and genomics

Roehrig et al., describe an integrated evaluation of 6 HB samples, 2 from the same tumor, using WGS, snRNA-Seq, and snATAC-Seq. They predict cell types, including cancer cell types, based on transcriptome clustering, identifying Hepatocytic, Liver Progenitor, and Mesenchymal HB cells. In previous work, Liver Progenitor HBs were identified as higher-risk tumors (Hirsch et al., 2021). The authors showed that heterogeneous tumors that contain multiple cancer subtypes differ in the location of these cell subtypes, and that these are spatially located in selected areas containing specific subtypes. Most interestingly, the authors show that chromatin remodeling is associated with the transcriptional programs of each of the three subtypes and identify TFs whose predicted binding sites are enriched in accessible regions in each subtype, and whose expression and regulons are differentially regulated in these subtypes.

One limitation of the study is the oversimplification of the histological HB subtypes included in the analysis, including two epithelial subtypes (fetal and embryonal) and mesenchymal, and 3 “poles” of differentiation. HBs are never pure mesenchymal tumors, but mixed epithelial and mesenchymal, and often demonstrate a wide variety of lineage differentiation, and undifferentiated (such as small cell or blastema) histological components. There is no sufficient histological annotation provided for the 6 samples profiled in the study, or sufficient clinical correlation provided. The illustration of the spatial transcriptomics analysis of the only tumor (Figure 3) could be greatly improved. If all these 6 HB samples were obtained post-chemotherapy, how do authors account for clonal selection and heterogeneity, and what were the areas selected for profiling? One of the cases illustrated in figure 1b (#3662T) from an older patient with a chemoresistant tumor carrying a TERT promoter mutation, clusters as C1 Cairo group (associated with fetal histology), while TERT promoter mutations are exclusively seen in aggressive tumors in the HCN-NOS histological group. On the other hand, the only patient that relapsed and died of disease was a young patient in the M (not LP) group carrying only a CTNNB1 variant.

As it stands, this approach appears excellent to investigate the differential expression and drivers of the previously reported Hepatocytic, Liver Progenitor, and Mesenchymal HB cells, based on the bulk RNA-seq transcriptomic groups. However, the main contribution of the manuscript is the association of promoter accessibility and the expression of Hirsch et al., gene signatures of the three subtypes, some reportedly associated with cisplatin resistance and prognosis. The identification of master regulators of each HB subtype relies on computational prediction alone, and the motif+regulon evidence is largely circular because it relies on previously identified interaction data—i.e., regulon identities and TF binding sites are not independent evidence. Moreover, the association of the 3 subtypes and chemoresistance relies wholly on previous publications by the authors. Thus, without additional evidence to support the identification of the gene regulatory networks associated with the 3 HB subtypes, the main thesis of this paper remains debatable and not supported. We would like to lobby the authors to curate additional

evidence through independent analysis and additional experiments. Without these data, the paper remains incomplete in its current form.

Reviewer #3 (Remarks to the Author): Expert in single-cell and spatial multi-omics, gene regulatory networks, cancer genomics, and bioinformatics

Summary:

In this manuscript, Roehrig et al. utilize whole genome sequencing and scMulti-Omics to study the genetic and epigenetic basis for tumor cell state heterogeneity in hepatoblastoma. Their results suggest that the prior gene expression classification may represent an “average” of an otherwise spectrum of cell states in each sample. Their work identifies a continuum of cell states between LP and H, with a hybrid state in between, as well as a distinct M state. In this paper, they show that heterogeneity within samples between these states may have a subclonal-genetic/epigenetic basis. This study represents an in-depth exploration of the molecular biology of hepatoblastoma tumor cells. However, the application of these results is limited without testing in a larger cohort. Likewise, while much of the bioinformatic approaches are sound, some convenient choices lack a purely empirical threshold or consideration. In addition, further explanations regarding data resources, sample size issues, and the integration analysis among multiple samples are necessary for the study.

Major Comments:

1. First and foremost, this dataset is very limited. The exhaustive nature of this dataset is not forgotten, but very few mechanistic insights can be obtained when the dataset is unable to fully cover the population variance. In addition, did the author use previously produced data or generate new data in this study? The authors should provide data generative type and technology in supplementary table 1 (named ST1 Multiome cohort in the Excel file).
2. Similarly, the author should also provide statistical power analysis for single-cell multi-ome (six samples across three conditions) and spatial cases (one sample) to secure the statistical rigor.
3. Fig. 1c visualization and insight are confused. Why would the author show the batches in the UMAP, and what information can be derived from this UMAP? Usually, the UMAP should visualize the post-integration cell cluster and show the consistent cell type across multiple conditions.
4. The authors likely overinterpret the PCA findings in Fig. 2. There are comments on PCA revealing two main axes of variation, yet no scree plot is provided to support such a claim. While PC1 and PC2 will certainly contain more variation than PC3 and beyond, most scRNA-Seq datasets will appropriately contain the most variance in at least 10 PCs. The only differentiable axis in Fig. 2A is PC1, but the dataset

only includes one mesenchymal sample, so there is no way of knowing if this is an outlier result or a real distinct population.

5. This manuscript could have really benefitted from an external look at bulk RNA-Seq or ATAC-Seq data to identify if the trends seen in the limited-sample single-cell data are present in a larger sample. The authors attempted this on a small cohort but excluded mesenchymal samples, which exist in the center of the scLP/scH continuum in PC2.

6. Regarding cell population identification, how did the author solve the integration issue among multiple samples? I cannot find the description for the integration analysis. PCA cannot solve the integration and identify consistent cell types based on multiple samples. I highly recommend that the author redo this analysis using integration tools like Seurat, Harmony, or scVI.

7. For the spatial analysis, the deconvolution analysis (e.g., using cell2location) is necessary to investigate cell composition in three regions, including fetal, embryonal, and intersection regions.

8. When selecting the TF target modules in Fig. 5, how do you justify selecting four modules naturally? The dendrogram at the top shows there's more variability within the scLP targets than there is between sc-epi and scH targets. It can be quite convenient to see roughly the results you're looking for, but if you split one with less variation, then you need to split the other. This is pretty clear in the Fig. 5b plots where scH/LP and scH do not really differentiate much from each other.

Minor Comments:

1. For the conclusion of Supplementary Fig. 3, please provide a quantitative description to showcase the consistency between WGS data and single-cell inferred copy number profiles (such as correlation).

2. There is very little justification for the four populations in Fig. 2. The gene sets clearly denote the three cell states previously identified. Beyond Fig. 2, the authors do not utilize this hybrid group much, nor do they explain them any further.

3. In Fig. 3, it would have been helpful to show scH and scLP markers on the zoomed-in region 1 and 2 to confirm the conclusions. It looks as though these markers are not that great at differentiating pathologist definitions on one of the two axes, which weakens the statements made. This is especially true for region 2, where scH markers essentially cannot differentiate between Embryonal and Fetal populations.

4. In the result, section of "Hepatoblastoma cells display continuous states along two differentiation axes," UMAP and t-SNE is not cell cluster identification method. It only provides visualization for cellular homogeneity and heterogeneity.

5. What is the mosaic cell? Please define it.

6. Provide a detailed figure legend in Fig. 1c.

Reviewer #4 (Remarks to the Author): Expert in single-cell multi-omics, spatial transcriptomics, gene regulatory networks, cancer genomics, and bioinformatics

Overview:

The manuscript (ms) presents a comprehensive analysis of hepatoblastoma cell transcriptomes, epigenomic landscapes, and somatic alterations at the single-cell level. The study successfully identifies differentiation between hepatoblastoma cell states (Hepatocytic, Liver Progenitor, H/LP intermediate, and Mesenchymal) and correlates them with prenatal liver development signatures. Additionally, the investigation of epigenetic remodeling, gene regulatory networks, and the interplay of clonal evolution and phenotypic plasticity provides valuable insights into hepatoblastoma progression and response to chemotherapy. Overall, this ms is well-written, the comprehensive data analysis results are well-presented, and this study significantly contributes to our understanding of hepatoblastoma biology. Nevertheless, I have a few comments that, if addressed, would significantly enhance the strength of the ms.

Comments:

1. In this study, it is mentioned that six hepatoblastoma (HB) samples were selected from the Hirsch data set, which contains 100 bulk RNA-seq samples. It would be helpful if the authors could provide more information on how these specific six samples were chosen. Was the selection random, or were there specific criteria used to ensure representativeness of the samples? Clarifying the sample selection process would enhance the transparency and reproducibility of the study.
2. Given the high variability and heterogeneity of human samples, it is crucial to address the potential batch (patient) effect when presenting the six snRNA-seq samples together on UMAP. However, the manuscript does not mention any data integration step or correction for the patient-specific effect. It would be valuable if the authors could discuss how they accounted for this strong patient-specific effect in the analysis. Describing any normalization or batch correction methods employed would strengthen the robustness of the results and ensure that the observed patterns are not solely driven by patient-specific differences.
3. In the visium spatial data analysis, it is stated that spots containing a mixture of tumor and stromal cells or hematopoiesis were excluded. It would be helpful if the authors could elaborate on how the cell type mixture was identified and distinguished from pure tumor or stromal cells. Providing details about

the criteria or markers used to determine the cell type composition of each spot would enhance the clarity and reliability of the spatial analysis results.

Reviewer #1 (Remarks to the Author): Expert in hepatoblastoma clinical research and genomics

This paper presented the interplay of clonal evolution and epigenetic plasticity shapes and suspected the potential of HB subclones to respond to chemotherapy using mainly single cell sequencing of HB tissues. The data are very interesting and integrated but the data seems as the make-up examination of the previous report (ref. 5) by the authors (Hirsh T.Z. et al. Cancer Discov. 2021). If authors want to submit this paper, the difference between this paper and the previous one by Hirsh T.Z et al. should be clarified and the followings should be solved or reconsidered. This previous paper already reported the scSeq and the classification of scH, scLP, scM, and scH/LP.

We warmly thank the Reviewer for his/her positive assessment of our manuscript and fruitful comments that helped us to improve the study in the revised version.

In our previous work (Hirsch *et al.*, Cancer Discov 2021), we had conducted an integrated genomic analysis of 100 bulk HB samples, and defined the 'hepatocytic' (H), 'liver progenitor' (LP) and 'mesenchymal' (M) molecular subgroups. By analyzing a single patient with single-nucleus RNA-seq, we had confirmed the coexistence of HB cells with different differentiation states.

The present work brings several important new insights compared with the previous one:

(i) By analyzing 21,150 cells from 6 carefully selected HB, representative of HB molecular and histological diversity (see answer to comment #1), we could explore in-depth the diversity of HB cell states. We show that tumor cells display a continuum of differentiation states between 3 poles. In particular, we identify the intermediate scH/LP cell with moderate expression of both scLP and scH markers.

(ii) By analyzing for the first time the chromatin accessibility landscape of HB cells, we unraveled the gene regulatory networks of each cell state, and the succession of transcription factors underlying cell state transitions. In the revision, we have strengthened this part by analyzing public ChIP-seq data for the major transcription factors associated with HB cell states. As suggested in your comment #6, we also examined the interplay between chromatin accessibility and methylation, revealing important chromatin remodeling of key imprinted regions.

(iii) Finally, we developed a strategy to map genetic subclones in single-cell data, which allowed us to reconstruct the plasticity of differentiation states in each subclone. We showed that each subclone displays its own range of plasticity, associated with its ability to proliferate after chemotherapy.

As requested, we have clarified the novelties of this work compared with the previous one in the revised Discussion section.

If the authors want to show “the clonal evolution and cellular plasticity in hepatoblastoma tissues”. The difference of scSEQ and histological examination. In this paper, the results of single-cell multiomics only show the same finding of histology.

The single-cell molecular profiles that we identify are indeed consistent with histology, but they allow to unravel the molecular determinants of cell identities, including the key transcription factors and gene regulatory networks governing each cell state. In addition, integrating the clonal evolution reveals how HB cell states evolve during tumor evolution. These data provide key insights to understand the histological heterogeneity of HB, and we have emphasized this point in the revised **Discussion** section.

1. As mentioned in their previous paper, ‘Liver Progenitor’ cells accumulated massive loads of cisplatin-induced mutations with a specific mutational signature, leading to the development of heavily mutated relapses and metastases. Therefore, genomic alteration will be induced by chemotherapy such as CDDP and Doxorubicin. The authors should clarify the reason why these 6 samples obtained after preoperative CTX were used.

We selected these 6 cases because they represented the diversity of HB molecular groups based on their bulk RNA-seq profiles. The principal component analysis (**Fig. 1a**) and the hierarchical clustering (see below) show that 3 samples belong to the 'Liver Progenitor', 2 to the 'Hepatocytic' and 1 to the 'Mesenchymal' subgroup. Thus, this selection allows us to cover the 3 main molecular subtypes identified in bulk RNA-seq. In addition, we have added in **Sup Fig. 2** the pathological reviewing of the

mirror blocks, indicating the proportions of fetal, embryonal and mesenchymal cells. Some samples were pure (e.g. #3610T, 100% mesenchymal) whereas others contained mixtures of histological components (e.g. #2959T, 70% embryonal and 30% fetal). For one patient, we selected two regions: one with a dominance of embryonal cells classified 'Liver Progenitor' in bulk RNA-seq, the other with a dominance of fetal cells classified 'Hepatocytic' in bulk RNA-seq. Overall, this selection allowed us to cover the molecular and histological diversity of HB, and to investigate both inter- and intra-sample heterogeneity. In addition, analyzing post-chemotherapy samples allowed us to explore the early response of subclones to treatment, as a function of the clonal evolution and differentiation states.

As noted by the Reviewer, 'Liver Progenitor' cells accumulate cisplatin-induced mutations during chemotherapy. These mutations are mostly subclonal passenger mutations, and we (Hirsch, Cancer Discov 2021) and others (Sekiguchi, NPG 2020) found similar driver mutation landscapes in pre- and post-chemotherapy samples. Thus, we do not expect chemotherapy-induced mutations to alter significantly the molecular profiles of HB. Consistently, pre- and post-chemotherapy samples are intermingled in our unsupervised classification (figure below), and distributed in the different molecular subgroups.

In the revised manuscript, we have added the new **Sup Fig. 1** showing where the 6 snMultiome cases, and pre / post-chemotherapy samples, are distributed in our unsupervised classification of 100 HB bulk RNA-seq profiles. We have also discussed the selection of the 6 samples in the first paragraph of the **Results** section.

Unsupervised clustering of 100 HB and 4 nontumor liver samples (bulk RNA-seq, Hirsch 2021 series)

New Supplementary Fig. 1. Transcriptomic classification of the 6 samples selected for single-nucleus Multiome. The dendrogram represents the hierarchical clustering of bulk RNA-seq profiles of 100 HB and 4 non-tumor liver samples (Hirsch data set). The 6 samples selected for snMultiome are indicated below, together with bulk molecular groups from the original study and sample type (pre / post-chemotherapy).

2. Moreover, why were these 6 cases selected from 100 cases including 18 tumors obtained at diagnosis? The authors should select the cases whose tumors were resected before chemotherapy.

As mentioned above, the driver alteration and bulk RNA-seq profiles were comparable between pre- and post-chemotherapy samples in our series. However, we fully agree that it is important to verify the relevance of our findings in pre-chemotherapy samples. All patients in France receive chemotherapy

before surgery, so the surgical resections we have are all obtained after chemotherapy. Before chemotherapy, we only have access to biopsies, and we do not have enough material for multiomic single-cell analyses. For example, the 18 pre-treatment biopsies used in our previous paper (Hirsch 2021, Cancer Discov) and mentioned by the Reviewer were entirely used for WGS and bulk RNA-seq, and we do not have material left to perform single-cell Multiome analyses from the same samples. Thus, in the revision, we used external data sets to validate our findings in 5 bulk RNA-seq data sets (314 HB samples in total) comprising pre-chemotherapy HB (Nagae 2021) post-chemotherapy HB (Carrillo-Reixach, JHEP2020; Hooks, Hepatology 2018) or both (Sekiguchi, NPJ 2020; Hirsch, Cancer Discov 2021). We also requested access to single-cell RNA-seq data from 13 treatment-naive HB (Wang, Cell Rep Med 2023) but we had no reply after an official data access request and 3 e-mails sent to corresponding authors.

We used Bisque tool (Jew, Nat Commun 2020) to estimate the proportion of scH, scH/LP, scLP and scM cells in bulk RNA-seq profiles. We first verified, in our 6 samples, that the proportions estimated from bulk were consistent with single-cell data (see below). We next applied Bisque to deconvolute the bulk RNA-seq profiles of all samples in our meta-series. Most samples comprised a mixture of several HB cell states, but their proportions were consistent with the bulk molecular groups and histological annotations of the samples. Besides, the estimated proportion of each cell state was strongly correlated with its GRN activity. Finally, we used the two series comprising both pre- and post-chemotherapy samples (Sekiguchi 2020 & Hirsch 2021) to compare the proportions of stromal and tumor cells. We found an overall decrease of tumor cells ($q = 0.028$) and increase of macrophages ($q = 5.5e-5$) after chemotherapy. The proportions of the 4 HB cell states did not differ significantly ($q > 0.05$). These new analyses support the robustness of HB cell states and GRNs in pre- and post-chemotherapy data sets. They have been added in the new paragraph "Deconvolution of HB cell states in bulk RNA-seq data" of the Results section, **Fig. 7** and **Sup Figs. 13** and **14**.

New Figure 7. Deconvolution of hepatoblastoma cell states in 100 bulk RNA-seq profiles. a Comparison of cell proportions identified in single-nucleus data or our 6 HB samples (left) and those estimated by Bisque deconvolution tool in matched bulk RNA-seq profiles (right). **b** Deconvolution of HB cell states from bulk RNA-seq profiles of 100 HB samples (Hirsch data set⁵). Two barplots indicate the % of stromal cells (top) and tumor cell states (middle) in each sample. Bulk transcriptomic subgroups and histological components identified in mirror blocks are annotated below. **c** Correlation between the proportions of scH, scLP and scM cells estimated by Bisque and the expression of their GRNs.

New Supplementary Fig. 13. Deconvolution of hepatoblastoma cell states in external data sets. a Deconvolution of HB cell states from bulk RNA-seq profiles of 214 HBs from 4 published series. Two barplots indicate the % of stromal cells (top) and tumor cell states (middle) in each sample. Samples are ordered according to their bulk transcriptomic subgroups. **b** Correlation between the proportions of scH, scLP and scM cells estimated by Bisque and the expression of their GRNs in each data set.

New Supplementary Fig. 14. Comparison of deconvoluted cell proportions between pre- and post-chemotherapy samples. We used bulk RNA-seq from two published series comprising both pre- and post-chemotherapy samples (Sekiguchi, *NPJ* 2020 & Hirsch, *Cancer Discov* 2021) to compare the abundance of stromal and tumor cells. Stromal and HB cell states were deconvoluted using Bisque tool. The proportions of each cell type were compared between 51 pre- and 99 post-chemotherapy samples using Wilcoxon's rank sum test, with Bonferroni correction for multiple testing.

3. Moreover, why did the authors examine only 2 non-tumorous liver tissues. The LOH analysis using WGS might require the individual normal tissue. Hirsch cohort had 18 normal tissues. The authors should add more data of non-tumorous liver tissues.

Sorry if this was not clearly explained in the manuscript but we did sequence the normal liver tissue of the 6 cases by WGS. This was necessary, as noted by the Reviewer, to identify germline SNPs and call LOH. We have rephrased the "Clinical samples" paragraph of the **Methods** section to clarify this point. In addition, two of the normal tissues were analyzed by single-nucleus Multiome to obtain the baseline expression and ATAC profiles of non-tumor liver. The analysis of more non-tumorous liver tissues is the topic of another article that we have in press at Nature Communications, by Pilet *et al.*, describing

the transcriptional landscape of non-tumor liver samples with or without mosaic 11p15 cnLOH in hepatoblastoma patients.

In addition

4. How did you detect “cnLOH” without the data of normal tissue?

As explained above, we did perform WGS of normal tissues. This has been clarified in the the "Clinical samples" paragraph of the **Methods** section.

5. In Figure 3, there are fibrosis and necrotic findings. How did you confirm the viable tumor cells in this specimen? It is possible that the scSeq showed only viability of the tumors

Necrotic, vascular, haemorrhagic and fibrotic areas were annotated in mirror blocks by an expert pathologist (see new **Sup. Fig. 2**) to ensure that the 6 samples were sufficiently rich in tumor cells to conduct genomic analyses. Stringent filters on the number of genes, UMIs and mitochondrial reads were applied to retain only viable cells in single-cell data. Quality control metrics of sample #3133T (included in **Fig. 3**) were good and in line with the rest of the series (**Sup Table 2** and **Sup Fig. 2**).

6. In ATAC seq data, the authors should describe its correlation with methylation status because the methylation of imprinting genes are well-known correlated to hepatoblastoma development.

We warmly thank the Reviewer for this suggestion, which allowed us to highlight key regulatory peaks in imprinted regions and include important new results in the revised manuscript.

First, we examined the correlation between ATAC-seq peak accessibility and DNA methylation, using our cohort of 84 HB samples analyzed by reduced-representation bisulfite sequencing (Hirsch, Cancer Discov 2021). Within our 6 samples with single-cell Multiome and RRBS data, we found a significant correlation between DNA methylation and the accessibility of overlapping peaks (Figure below, panel a), with a majority of accessible peaks overlapping demethylated regions (methylation < 10%).

Second, we compared the chromatin accessibility signatures of our scM, scLP and scH cells with our previously described methylation components associated with mesenchymal, liver progenitor and hepatocytic subtypes. We found a significant anti-correlation between DNA accessibility and DNA methylation, both for regions distinguishing mesenchymal from epithelial differentiation, and for regions distinguishing liver progenitor from hepatocytic differentiation (Figure below, panel b).

These results refine our understanding of the relationship between chromatin accessibility and DNA methylation in HB. They have been added in the new **Sup Fig. 10**.

New Supplementary Fig. 10. Correlation between chromatin accessibility and DNA methylation. a Boxplot showing the distribution of chromatin accessibility in snATAC-seq as a function of the methylation of overlapping 100 bp tiles in reduced representation bisulfite sequencing (RRBS) data (Hirsch et al., *Cancer Discov* 2021). **b** Correlation between the contribution of 100 bp tiles to DNA methylation components distinguishing HB subtypes and the chromatin accessibility changes observed between the same subtypes. Left: Mesenchymal vs. epithelial. Right: Hepatocytic vs. Liver Progenitor.

Finally, as indicated by the Reviewer, the deregulation of several imprinted regions is implicated in HB development, including the *IGF2* locus at 11p15, and the *DLK1/MEG3* locus at 14q32. We examined our ATAC-seq data to identify chromatin accessibility changes in these imprinted regions.

Among the 108 ATAC-seq peaks located within the imprinted 11p15 region, 52 (resp. 13) were significantly more (resp. less) accessible in HB cells as compared with normal liver cells. The 52 peaks with increased accessibility in HB included 8 peaks linked with *IGF2* expression, and 10 with *KCNQ1OT1* expression, 2 genes expressed from the paternal allele and activated in HB by duplication of the paternal allele (present in 5/6 tumors of this series), gain of methylation of imprinted control

region 1 (GOM IC1) or loss of methylation at imprinted control region 2 (LOM IC2). These deregulated peaks do not overlap with the IC1/2 region, but they highlight important regulatory regions in the locus, including 8 that control the expression of *IGF2*, a key oncogene in HB.

The imprinted *DLK1/MEG3* locus at 14q32 has been shown to be overexpressed in HB and associated with poor outcome (Carrillo-Reixach *et al.*, JHEP 2020). Consistently, 32/40 ATAC-seq peaks in this region were significantly more accessible in HB cells as compared with normal liver cells. These comprise peaks linked with the expression of several genes included in the bad prognosis "14q32 signature" of Carrillo-Reixach, such as *DLK1*, *MEG3*, *RTL1*, *MEG8* and *MEG9*.

Altogether, this analysis revealed differential accessibility in 2 imprinted regions known to be involved in HB development, and highlighted regulatory peaks of major oncogenes like *IGF2* and several genes of the *DLK1/MEG3* locus. They have been added in the new **Sup Table 6** and **Sup Fig. 9** (below).

New Supplementary Fig. 9. Differential chromatin accessibility in imprinted regions. Shown are 4 regions located in imprinted loci known to be implicated in HB development (11p15 and 14q32), with significant differentially accessible ATAC-seq peaks between HB and non-tumor (NT) liver cells. The

first two tracks represent the normalized snATAC-seq signal in HB and NT cells. The two tracks below represent the methylation levels identified by bulk reduced representation bisulfite sequencing in Hirsch data set.

7. In Fig 4a, what is “N”. Non-tumor samples were only 2. If you said that “N” is non-tumor hepatocytes in the tumor tissue, you have to decline how to detect them.

In **Fig. 4a**, NT correspond to non-tumor hepatocytes in the two non-tumor samples (n=1,518). We have now indicated the number of cells used for the differential accessibility analysis in the Figure Legend.

8. Each figures are too small to be understood. The authors should reproduce them to be understood easily.

In the revision, we have increased the size of the smallest panels to improve their readability. We also increased font sizes to meet the requirements of Nature Communications.

Reviewer #2 (Remarks to the Author): Expert in hepatoblastoma clinical research and genomics

Roehrig et al., describe an integrated evaluation of 6 HB samples, 2 from the same tumor, using WGS, snRNA-Seq, and snATAC-Seq. They predict cell types, including cancer cell types, based on transcriptome clustering, identifying Hepatocytic, Liver Progenitor, and Mesenchymal HB cells. In previous work, Liver Progenitor HBs were identified as higher-risk tumors (Hirsch et al., 2021). The authors showed that heterogeneous tumors that contain multiple cancer subtypes differ in the location of these cell subtypes, and that these are spatially located in selected areas containing specific subtypes. Most interestingly, the authors show that chromatin remodeling is associated with the transcriptional programs of each of the three subtypes and identify TFs whose predicted binding sites are enriched in accessible regions in each subtype, and whose expression and regulons are differentially regulated in these subtypes.

We thank the Reviewer for his/her positive evaluation of our work. We have addressed below all the comments raised, which helped us significantly strengthen the messages of the study.

One limitation of the study is the oversimplification of the histological HB subtypes included in the analysis, including two epithelial subtypes (fetal and embryonal) and mesenchymal, and 3 “poles” of differentiation. HBs are never pure mesenchymal tumors, but mixed epithelial and mesenchymal, and often demonstrate a wide variety of lineage differentiation, and undifferentiated (such as small cell or blastema) histological components. There is no sufficient histological annotation provided for the 6 samples profiled in the study, or sufficient clinical correlation provided.

We agree and we have now provided in Sup Table 1 the detailed reviewing of mirror blocks by an expert pathologist (Dr Guillaume Morcrette), including the proportions of each component. The reviewing slides are also provided in the new **Sup Fig. 2** (see below). We had carefully selected our 6 samples based on histology and the molecular group of matched bulk RNA-seq data (shown in the new **Sup Fig. 1**), to be sure to include cells from the different contingents. As noted by the Reviewer, HB are never purely mesenchymal. The tumor from which sample #3610T was taken also contained epithelial contingents. However, the mirror block of the sample used in this study contained 100% mesenchymal cells, consistent with our single-cell results. In the revision, we have better explained our sample selection in the first paragraph of the Results section.

Sample ID	Main histology	Detailed reviewing of the mirror block
#2959T	Embryonal	70% Embryonal intricated with 30% Fetal with some intermediate pattern / rosettes / No hematopoiesis / No inflammation
#2960T	Fetal	100% Fetal / pseudoacinous pattern + few bile plugs / lymphocyte infiltrate ++ / small arterioles within fibrotic spots / No hematopoiesis
#3133T	Embryonal	70% Embryonal intricated with 30% Fetal with some intermediate pattern / rosettes/ No hematopoiesis / No inflammation / 5% fibrosis
#3377T	Fetal	100% Fetal / Steatosis / pseudoacinous pattern + few bile plugs + little trabecular trend spot / lymphocytes macrophages siderophages infiltrate + / 2 hematopoietic spots (erythroblastic only) / few arterioles / Fibrosis 15%
#3610T	Mesenchymal	100% Mesenchymal: 80% immature mesenchyma + 20% osteoid / No hematopoiesis / No or scarce inflammation
#3662T	Fetal	60 % fetal with some vascular bloody spaces, few pseudoacinous pattern + rare bile plugs / 40% Embryonal with rares rosettes intricated with a little Fetal with some intermediate pattern +/- trabecular / 40% fibrosis, thrombosed vessels, siderophages++ lymphocytes+ few calcifications / No obvious hematopoiesis

Detailed reviewing of the mirror blocks added in Sup Table 1

#2959T

FFPE section / mirror block / 100% tumoral
70% Embryonal component intricicated with 30% Fetal component with some intermediate
pattern / rosettes / No hematopoiesis / No inflammation.

#2960T

FFPE section / mirror block /
100% Tumoral Fetal component / pseudoacinar pattern + few bile plugs /
Lymphocyte infiltrate ++ / small arterioles within fibrotic spots / No hematopoiesis.

#3133T

FFPE section / mirror block /
70% Embryonal intricated with 30% Fetal with some intermediate pattern / rosettes /
No hematopoiesis / No inflammation / 5% fibrosis.

#3377T

FFPE section / mirror block /
100% Tumoral Fetal component

HESx02

100% Tumoral Fetal component
Steatosis / pseudoacinar pattern + few bile plugs + little trabecular trend spots /
Lymphocytes macrophages siderophages infiltrate + / 2 hematopoietic spots / few arterioles

HESx20

#3610T

Frozen section / Hamamatsu C13210 digitalization x40
100% Tumoral mesenchymal component (80% immature mesenchyma + 20% osteoid)
No hematopoiesis / No or scarce inflammation.

#3662T

FFPE section / mirror block /
60 % tumor / 40% fibrosis

Fetal with some vascular bloody spaces, few pseudoacinous pattern + rare bile plugs /
Embryonal with rares rosettes intricated with a little Fetal with some intermediate pattern +/-
trabecular / thrombosed vessels, siderophages++ lymphocytes+ few calcifications / No obvious
hematopoiesis.

New Supplementary Fig. 2. Mirror block reviewing of the 6 cases analyzed by single-nucleus Multiome.

The illustration of the spatial transcriptomics analysis of the only tumor (Figure 3) could be greatly improved.

We agree and we have completed this analysis in the revised manuscript. As suggested by Reviewer #3, we have added the expression of scH and scLP markers in the zoomed-in regions 1 and 2 in the revised **Fig. 3** (see below). In addition, we used a computational tool (SPOTlight, Elosua-Bayes *et al.*, NAR 2021) to deconvolute HB cell states in Visium spots. Cell composition was associated with histological annotations, with an enrichment of scLP cells in embryonal spots and scH cells in fetal spots. ScH/LP cells were quite abundant, although this population is heterogeneous and may include a range of phenotypes between scLP and scH. We have added this analysis in the Results section and in the new **Fig. 3b** and **Supplementary Fig. 8** (see below).

Revised Fig. 3. Spatial transcriptomics analysis of tumor #3133T. a Hematoxylin and eosin staining in an FFPE section of tumor #3133T. The two annotated regions correspond to embryonal nodules

surrounded by fetal areas. **b** Ternary graph showing the deconvoluted proportions of epithelial HB cell states in each Visium spot of regions 1 (left) and 2 (right). Visium spots were annotated by a pathologist and are colored by histology. **c** Focus on region 1 of the FFPE slide with a nodule of embryonal cells surrounded by fetal cells. **d** Spatial distribution of the mean expression of scH and scLP markers (Supplementary Table 3) in region 1. **e** Anti-correlation of scH and scLP markers across Visium spots in region 1, colored according to their histological annotation in panel c. **f** Focus on region 2 of the FFPE slide, with spots annotated by histology. **g** Spatial distribution of the mean expression of scH and scLP markers in region 2. **h** Anti-correlation of scH and scLP markers across Visium spots in region 2, colored according to their histological annotation in panel f.

New Supplementary Fig. 8. Deconvolution of HB cell states in spatial transcriptomics data. a Deconvolution of HB cell states in region 1. The upper left panel shows the histological annotations of spots. The upper right panel shows a zoomed-in focus on H&E staining of a region at the interface between fetal and embryonal cells. The bottom panel shows the results of the deconvolution, with a pie chart showing the estimated cell state proportions in each spot. **b** Deconvolution of HB cell states in region 2.

If all these 6 HB samples were obtained post-chemotherapy, how do authors account for clonal selection and heterogeneity, and what were the areas selected for profiling?

The 6 samples analyzed by single-nucleus Multiome were carefully selected to account for the diversity of HB, based on the histological evaluation of their mirror blocks (see above), and on their molecular signatures in matched bulk RNA-seq data. The hierarchical clustering below (100 bulk RNA-seq from Hirsch *et al.*, new **Sup. Fig. 1**) show that 3 samples belong to the 'Liver Progenitor', 2 to the 'Hepatocytic' and 1 to the 'Mesenchymal' subgroup. Thus, this selection covers the 3 main bulk molecular subtypes, and the different cell morphologies observed in HB.

Unsupervised clustering of 100 HB and 4 nontumor liver samples (bulk RNA-seq, Hirsch 2021 series)

New Supplementary Fig. 1. Transcriptomic classification of the 6 samples selected for single-nucleus Multiome. The dendrogram represents the hierarchical clustering of bulk RNA-seq profiles of 100 HB and 4 non-tumor liver samples (Hirsch data set). The 6 samples selected for snMultiome are indicated below, together with bulk molecular groups from the original study and sample type (pre / post-chemotherapy).

Using a mutational signature of cisplatin, we previously showed that chemotherapy-induced mutations are mostly subclonal passenger mutations. We (Hirsch, *Cancer Discov* 2021) and others (Sekiguchi, *NPG* 2020) found similar driver mutation landscapes in pre- and post-chemotherapy samples. Thus, we do not expect chemotherapy-induced mutations to alter significantly the molecular profiles of HB. Consistently, pre- and post-chemotherapy samples are intermingled in our unsupervised classification (figure above), and distributed in the different molecular subgroups.

However, we fully agree that it is important to verify the relevance of our findings in pre-chemotherapy samples. In France, all patients receive chemotherapy before surgery, so the surgical resections we have are all obtained after chemotherapy. Before chemotherapy, we only have access to biopsies, and we do not have enough material for multiomic single-cell analyses. Thus, in the revision, we used external data sets to validate our findings in 5 bulk RNA-seq data sets (314 HB samples in total) comprising pre-chemotherapy HB (Nagae 2021) post-chemotherapy HB (Carrillo-Reixach, *JHEP*2020; Hooks, *Hepatology* 2018) or both (Sekiguchi, *NPJ* 2020; Hirsch, *Cancer Discov* 2021). We also requested access to single-cell RNA-seq data from 13 treatment-naive HB (Wang, *Cell Rep Med* 2023) but we had no reply after an official data access request and 3 e-mails sent to corresponding authors.

We used Bisque tool (Jew, Nat Commun 2020) to estimate the proportions of scH, scH/LP, scLP and scM cells in bulk RNA-seq profiles. We first verified, in our 6 samples, that the proportions estimated from bulk were consistent with single-cell data (see below).

New Figure 7a. Deconvolution of hepatoblastoma cell states in bulk RNA-seq profiles. Comparison of cell proportions identified in single-nucleus data or our 6 HB samples (left) and those estimated by Bisque deconvolution tool in matched bulk RNA-seq profiles (right).

We next applied Bisque to deconvolute the bulk RNA-seq profiles of all samples in our meta-series. Most samples comprised a mixture of several HB cell states, but their proportions were consistent with the bulk molecular groups and histological annotations of the samples. Besides, the estimated proportion of each cell state was strongly correlated with its GRN activity (see answer to your last comment). Finally, we used the two series comprising both pre- and post-chemotherapy samples (Sekiguchi 2020 & Hirsch 2021) to compare the proportions of stromal and tumor cells. We found an overall decrease of tumor cells ($q = 0.028$) and increase of macrophages ($q = 5.5e-5$) after chemotherapy. The proportions of the 4 HB cell states did not differ significantly ($q > 0.05$). These results support the robustness of HB cell states and GRNs in pre- and post-chemotherapy data sets. They have been added in the new paragraph "Deconvolution of HB cell states in bulk RNA-seq data" of the Results section, and in **Sup Fig. 14** (below).

New Supplementary Fig. 14. Comparison of deconvoluted cell proportions between pre- and post-chemotherapy samples. We used bulk RNA-seq from two published series comprising both pre- and post-chemotherapy samples (Sekiguchi, *NPJ* 2020 & Hirsch, *Cancer Discov* 2021) to compare the abundance of stromal and tumor cells. Stromal and HB cell states were deconvoluted using Bisque tool. The proportions of each cell type were compared between 51 pre- and 99 post-chemotherapy samples using Wilcoxon's rank sum test, with Bonferonni correction for multiple testing.

One of the cases illustrated in figure 1b (#3662T) from an older patient with a chemoresistant tumor carrying a *TERT* promoter mutation, clusters as C1 Cairo group (associated with fetal histology), while *TERT* promoter mutations are exclusively seen in aggressive tumors in the HCN-NOS histological group. On the other hand, the only patient that relapsed and died of disease was a young patient in the M (not LP) group carrying only a *CTNNB1* variant.

Indeed, tumor #3662T is from a 3.8 yo patient with a chemoresistant tumor carrying a *TERT* promoter mutation. It is classified in C1 Cairo group (associated with fetal histology), but in the most aggressive LP Hirsch group (associated with embryonal histology). This is probably explained by the heterogeneity of this tumor, seen both in the histological annotation of the mirror block, and in single-nucleus data. Note that the most embryonal subclone 3, that represents only 25 cells in single-nucleus data (**Fig. 7f**),

is dominant in the sample used for bulk analyses, based on the clonality of its specific copy-number alterations +2p,8,12 and -20. This explains why the LP signature is stronger in bulk than single-nucleus RNA-seq for this tumor. In the revised Discussion section, we have added a comment regarding the interest of deconvoluting single cell states in bulk data to better characterize intra-sample heterogeneity missed by bulk molecular groups. Regarding *TERT*, we have an ongoing study on a large series showing that, in multivariate analysis, *TERT* promoter mutations are associated with age but not with molecular groups nor survival. #3662T is the youngest case with *TERT* promoter mutation in Hirsch series (100 HB). It has been reviewed as HB by two expert pathologists.

The only patient who relapsed and died (#3133T) was actually of the LP subgroup, and carried a *CTNNB1* S33Y mutation and a cn-LOH of 11p15 locus.

As it stands, this approach appears excellent to investigate the differential expression and drivers of the previously reported Hepatocytic, Liver Progenitor, and Mesenchymal HB cells, based on the bulk RNA-seq transcriptomic groups. However, the main contribution of the manuscript is the association of promoter accessibility and the expression of Hirsch et al., gene signatures of the three subtypes, some reportedly associated with cisplatin resistance and prognosis. The identification of master regulators of each HB subtype relies on computational prediction alone, and the motif+regulon evidence is largely circular because it relies on previously identified interaction data—i.e., regulon identities and TF binding sites are not independent evidence. Moreover, the association of the 3 subtypes and chemoresistance relies wholly on previous publications by the authors. Thus, without additional evidence to support the identification of the gene regulatory networks associated with the 3 HB subtypes, the main thesis of this paper remains debatable and not supported. We would like to lobby the authors to curate additional evidence through independent analysis and additional experiments. Without these data, the paper remains incomplete in its current form.

We thank the Reviewer for raising these weaknesses. In the revised manuscript, we leveraged independent data sets and performed additional analyses to strengthen the conclusions of the paper.

Validation of HB cell differentiation axes

First, we explored the diversity of HB cell states in an independent single-cell RNA-seq data set comprising 6,244 tumor cells from 9 HB (Song *et al.*, Nat Commun 2022). Like in our series, principal component analysis revealed 2 main axes of variation, with 3 poles characterized by a high expression of scH, scLP and scM markers (see figure below). As expected, scH markers were highly expressed in purely fetal samples, and scM markers were highly expressed in mixed epithelial-mesenchymal tumors. scLP markers were only expressed in few cells from a single sample, suggesting that the embryonal contingent is poorly represented in this data set. Unfortunately, mirror block histological annotations are not provided so the precise histological content of each sample is unknown. Altogether, these results are consistent with the cell states identified in our series. The scLP and the intermediate scH/LP states are better represented in our data set because we carefully selected fetal and embryonal samples based on mirror block histology and matched bulk RNA-seq. These results have been added in the main text and in the new **Sup Fig. 7**.

New Supplementary Fig. 7. UMAP and principal component analysis of Song's scRNA-seq data set. **a** UMAP visualization of 6,244 tumor cells from 9 HB show a grouping of cells by sample of origin. **b** Projection of HB cells on the first two principal components result in a mixture of cells from different samples. **c** Projection of HB cells on the first two principal components, with a color code showing the expression of scH, scLP and scM markers.

Validation of master regulators using ChIP-seq data

To identify the master regulators of each cell state, we had initially integrated RNA-seq and ATAC-seq data to identify the transcription factors (TFs) that (i) were up-regulated, (ii) had their binding motifs more accessible and (iii) had their target genes overexpressed in the relevant cell type. In our opinion, this evidence is not circular because RNA-seq and ATAC-seq provide independent evidence of the implications of the TFs. However, we agree that the binding motifs relied on computational prediction.

In the revised manuscript, we used ChIP-seq data from the ReMap database (<https://remap2022.univ-amu.fr>) to provide additional evidence supporting the master TFs of each cell state. The 4th release of ReMap (2022) compiles 8,103 quality controlled ChIP-seq datasets from various public sources such as GEO, ArrayExpress and ENCODE. We systematically analyzed the overlap between ATAC-seq peaks more accessible in each cell state and TF ChIP-seq peaks from ReMap. Although several TFs were not represented in ReMap (or not in a relevant cell type), we found a significant association for 6 TFs of the scH module (AR, CEBPB, CEBPD, POU5F1, HLF and NIL3), 7 TFs of the sc-epi module (HNF1A, HNF4A, HNF4G, ONECUT1, ONECUT2, FOXA1 and GATA4), 4 TFs of the scLP module (MAZ, MYCN, FOXA2 and BCL11A) and 4 TFs of the scM module (LEF1, RUNX2, TCF3 and ESR1). These TFs with ChIP-seq validation are now indicated in **Sup Table 7**. Representative TF-binding regions (see below) are shown in **Fig. 4d** and **Sup Fig. 11**.

d

New Fig. 4d showing representative examples of overlaps between ATAC-seq and ChIP-seq peaks of TFs associated with HB cell states (HNF1A for sc-epi, LEF1 for scM, CEBPB for scH and MAZ for scLP).

CEBPB (scH GRN)

Overlap of CEBPB ChIP-seq peaks with the 5,193 ATAC-seq peaks more accessible in scH: $q=4.5e-6$

MAZ (scLP GRN)

Overlap of MAZ ChIP-seq peaks with the 9,432 ATAC-seq peaks more accessible in scLP: $q=1.4e-5$

HNF1A (sc-epi GRN)

Overlap of HNF1A ChIP-seq peaks with the 28,836 ATAC-seq peaks more accessible in sc-epithelial: $q=3.0e-122$

LEF1 (scM GRN)

Overlap of LEF1 ChIP-seq peaks with the 20,010 ATAC-seq peaks more accessible in scM: $q=1.9e-15$

New Sup Fig. 11. Overlap between differentially accessible ATAC-seq peaks and transcription factor ChIP-seq peaks. Eight representative regions are shown, displaying the overlap between ChIP-seq peaks of 4 key transcription factors (HNF1A associated with sc-epithelial cells, LEF1 with scM cells, CEBPB with scH cells and MAZ with scLP cells), together with the normalized ATAC-seq signals in scH, scLP and scM cells. All significant associations are reported in **Sup Table 7**. ChIP-seq coverage tracks were obtained from ENCODE (HNF1A in HepG2 cells: ENCFF502ACF; CEBPB in HepG2 cells: ENCFF406BBU; MAZ in HepG2 cells: ENCFF527EZL) or GEO (LEF1 in hESC: GSM1579343).

Validation of GRNs in independent single-cell and bulk RNA-seq data

In the revised manuscript, we used published single-cell RNA-seq data (6,244 cells from 9 patients, Song, Nat Commun 2022) and bulk RNA-seq data (314 HB samples from 5 series: Carrillo-Reixach, JHEP2020; Hooks, Hepatology 2018; Sekiguchi, NPJ 2020; Nagae, Nat Commun 2021, Hirsch, Cancer Discov 2021) to validate the GRNs underlying each cell state and their association with bulk molecular groups and histological annotations.

First, we validated the modules of co-regulated transcription factors and targets that define each cell state. To that aim, we calculated the correlations between each TF / target belonging to one of the GRNs,

and we represented these correlations in a heatmap, with TFs and targets ordered as in our Fig. 5a (representing the GRNs in our snMultiome cohort). We performed this analysis at single-cell scale using Song's data set, and at bulk scale using the 5 series mentioned above. We obtained correlation patterns consistent with those observed in our data set, validating these co-expressed TF-target modules in a large HB meta-series (see below).

New Supplementary Figure 12. Validation of GRNs in external data sets. a The correlation of the GRN transcription factors (TFs) and target genes were validated in one single-cell RNA-seq data set and 5 bulk RNA-seq data sets totalling 314 HB samples.

We also used a deconvolution approach (Bisque, Jew *et al.* Nat Commun 2020) to estimate the proportions of the 4 HB cell states (scH, scH/LP, scLP and scM), and the 4 main types of stromal cells, from bulk RNA-seq data. We first tested Bisque on our 6 samples with matched snMultiome and bulk RNA-seq data, and we obtained consistent results (panel a below). We then applied Bisque to our entire series of 100 HB. The proportions of HB cell states were consistent with the bulk molecular group of each sample and with the histological components of the mirror block (panel b below). However, almost every sample comprised a mixture of several states. Interestingly, the estimated proportion of each cell state was strongly correlated with its GRN activity (panel c below). Similar results were obtained in the 4 other bulk RNA-seq series. These results support the robustness of HB cell states and GRNs, and suggest that deconvolution approaches may provide a more subtle characterization of bulk RNA-seq profiles.

These new analyses have been added in the new paragraph "Deconvolution of HB cell states in bulk RNA-seq data" of the Results section, **Fig. 7** and **Sup Figs. 12** and **13**.

New Figure 7. Deconvolution of hepatoblastoma cell states in 100 bulk RNA-seq profiles. *a* Comparison of cell proportions identified in single-nucleus data or our 6 HB samples (left) and those estimated by Bisque deconvolution tool in matched bulk RNA-seq profiles (right). *b* Deconvolution of HB cell states from bulk RNA-seq profiles of 100 HB samples (Hirsch data set⁵). Two barplots indicate the % of stromal cells (top) and tumor cell states (middle) in each sample. Bulk transcriptomic subgroups and histological components identified in mirror blocks are annotated below. *c* Correlation between the proportions of scH, scLP and scM cells estimated by Bisque and the expression of their GRNs.

New Supplementary Fig. 13. Deconvolution of hepatoblastoma cell states in external data sets. a Deconvolution of HB cell states from bulk RNA-seq profiles of 214 HBs from 4 published series. Two barplots indicate the % of stromal cells (top) and tumor cell states (middle) in each sample. Samples are ordered according to their bulk transcriptomic subgroups. **b** Correlation between the proportions of scH, scLP and scM cells estimated by Bisque and the expression of their GRNs in each data set.

Reviewer #3 (Remarks to the Author): Expert in single-cell and spatial multi-omics, gene regulatory networks, cancer genomics, and bioinformatics

Summary:

In this manuscript, Roehrig et al. utilize whole genome sequencing and scMulti-Omics to study the genetic and epigenetic basis for tumor cell state heterogeneity in hepatoblastoma. Their results suggest that the prior gene expression classification may represent an “average” of an otherwise spectrum of cell states in each sample. Their work identifies a continuum of cell states between LP and H, with a hybrid state in between, as well as a distinct M state. In this paper, they show that heterogeneity within samples between these states may have a subclonal-genetic/epigenetic basis. This study represents an in-depth exploration of the molecular biology of hepatoblastoma tumor cells. However, the application of these results is limited without testing in a larger cohort. Likewise, while much of the bioinformatic approaches are sound, some convenient choices lack a purely empirical threshold or consideration. In addition, further explanations regarding data resources, sample size issues, and the integration analysis among multiple samples are necessary for the study.

We warmly thank the Reviewer for his/her positive assessment of our work, and for the numerous suggestions that helped us considerably improve the manuscript.

Major Comments:

1. First and foremost, this dataset is very limited. The exhaustive nature of this dataset is not forgotten, but very few mechanistic insights can be obtained when the dataset is unable to fully cover the population variance. In addition, did the author use previously produced data or generate new data in this study? The authors should provide data generative type and technology in supplementary table 1 (named ST1 Multiome cohort in the Excel file).

The single-cell Multiome data was generated for this study. We have now indicated in **Sup Table 1**, as requested, the type of sequencing data that was generated for this study, or previously published.

We have selected for this study 6 HB that represent the diversity of HB, based on their bulk RNA-seq profiles. This was shown initially by principal component analysis (Fig. 1a). In the revised manuscript, we also projected these samples in a hierarchical clustering of 100 HB, showing that they represent the diversity of molecular subgroups: 3 samples belong to the 'Liver Progenitor', 2 to the 'Hepatocytic' and 1 to the 'Mesenchymal' subgroup (new **Sup. Fig. 1**, see below).

Unsupervised clustering of 100 HB and 4 nontumor liver samples (bulk RNA-seq, Hirsch 2021 series)

New Supplementary Fig. 1. Transcriptomic classification of the 6 samples selected for single-nucleus Multiome. The dendrogram represents the hierarchical clustering of bulk RNA-seq profiles of 100 HB

and 4 non-tumor liver samples (Hirsch data set). The 6 samples selected for snMultiome are indicated below, together with bulk molecular groups from the original study and sample type (pre / post-chemotherapy).

However, we agree that our findings need to be validated in a larger cohort. In the revised manuscript, we used published single-cell RNA-seq data (6,244 cells from 9 patients, Song, Nat Commun 2022) and bulk RNA-seq data (314 HB samples from 5 series: Carrillo-Reixach, JHEP2020; Hooks, Hepatology 2018; Sekiguchi, NPJ 2020; Nagae, Nat Commun 2021, Hirsch, Cancer Discov 2021) to validate (1) the diversity of HB cell states, (2) the gene regulatory networks underlying each cell state and (3) their association with bulk molecular groups and histological annotations.

First, we used Song's single-cell RNA-seq data to explore the diversity of HB cell states in an independent data set. Like in our series, the UMAP projection of tumor cells was strongly patient-driven, due to the specific copy-number changes of each tumor (see also answer to comment #6). By contrast, principal component analysis revealed 2 main axes of variation, with 3 poles characterized by a high expression of scH, scLP and scM markers (see figure below). As expected, scH markers were highly expressed in purely fetal samples, and scM markers were highly expressed in mixed epithelial-mesenchymal tumors. scLP markers were only expressed in few cells from a single sample, suggesting that the embryonal contingent is poorly represented in this data set. Unfortunately, mirror block histological annotations are not provided so the precise histological content of each sample is unknown. Altogether, these results are consistent with the cell states identified in our series. The scLP and the intermediate scH/LP states are better represented in our data set because we carefully selected fetal and embryonal samples based on mirror block histology and matched bulk RNA-seq. These results have been added in the main text and in **Sup Fig. 7**.

New Supplementary Fig. 7. UMAP and principal component analysis of Song's scRNA-seq data set. a UMAP visualization of 6,244 tumor cells from 9 HB show a grouping of cells by sample of origin. **b**

Projection of HB cells on the first two principal components result in a mixture of cells from different samples. c Projection of HB cells on the first two principal components, with a color code showing the expression of scH, scLP and scM markers.

Second, we validated the gene regulatory networks (GRNs), e.g. the modules of co-regulated transcription factors and targets that define each cell state. To that aim, we calculated the correlations between each TF / target belonging to one of the GRNs, and we represented these correlations in a heatmap, with TFs and targets ordered as in our Fig. 5a (representing the GRNs in our snMultiome cohort). We performed this analysis at single-cell scale using Song's data set, and at bulk scale using the 5 series mentioned above. We obtained correlation patterns consistent with those observed in our data set, validating these co-expressed TF-target modules in a large HB series (see below).

New Supplementary Figure 12. Validation of GRNs in external data sets. *a* The correlation of the GRN transcription factors (TFs) and target genes were validated in one single-cell RNA-seq data set and 5 bulk RNA-seq data sets totalling 314 HB samples.

Finally, we used a deconvolution approach (Bisque, Jew *et al.* Nat Commun 2020) to estimate the proportions of the 4 HB cell states (scH, scH/LP, scLP and scM), and the 4 main types of stromal cells, from bulk RNA-seq data. We first tested Bisque on our 6 samples with matched snMultiome and bulk RNA-seq data, and we obtained consistent results (panel a below). We then applied Bisque to our entire series of 100 HB. The proportions of HB cell states were consistent with the bulk molecular group of each sample and with the histological components of the mirror block (panel b below). However, almost every sample comprised a mixture of several states. Interestingly, the estimated proportion of each cell state was strongly correlated with its GRN activity (panel c below). Similar results were obtained in the 4 other bulk RNA-seq series. These results support the robustness of HB cell states and GRNs, and suggest that deconvolution approaches may provide a more subtle characterization of bulk RNA-seq profiles.

These new analyses have been added in the new paragraph "Deconvolution of HB cell states in bulk RNA-seq data" of the Results section, **Fig. 7** and **Sup Fig. 12** and **13**.

New Figure 7. Deconvolution of hepatoblastoma cell states in 100 bulk RNA-seq profiles. *a* Comparison of cell proportions identified in single-nucleus data or our 6 HB samples (left) and those estimated by Bisque deconvolution tool in matched bulk RNA-seq profiles (right). *b* Deconvolution of HB cell states from bulk RNA-seq profiles of 100 HB samples (Hirsch data set⁵). Two barplots indicate the % of stromal cells (top) and tumor cell states (middle) in each sample. Bulk transcriptomic subgroups and histological components identified in mirror blocks are annotated below. *c* Correlation between the proportions of scH, scLP and scM cells estimated by Bisque and the expression of their GRNs.

New Supplementary Fig. 13. Deconvolution of hepatoblastoma cell states in external data sets. a Deconvolution of HB cell states from bulk RNA-seq profiles of 218 HBs from 4 published series. Two barplots indicate the % of stromal cells (top) and tumor cell states (middle) in each sample. Samples are ordered according to their bulk transcriptomic subgroups. **b** Correlation between the proportions of scH, scLP and scM cells estimated by Bisque and the expression of their GRNs in each data set.

2. Similarly, the author should also provide statistical power analysis for single-cell multiome (six samples across three conditions) and spatial cases (one sample) to secure the statistical rigor.

In the revision, we performed a statistical power analysis using scPower tool (Schmid *et al.*, Nat Commun 2021). We estimated that, with the number of samples, cells per sample and reads per cell of our study, we have 95% power to identify a population representing 2.7% of all cells. Given their relative

prevalence in the data set, the detection power of differentially expressed genes was 0.63 for scH cells, 0.62 for scLP cells and 0.57 for scM cells. We have now indicated these numbers in the Methods section. The spatial transcriptomics was not used to discover cell types or gene expression signatures, but just to visualize the distribution of cell states in a representative case with two LP nodules. Thus, we do not see what power estimation can be made. However, we have indicated in the Discussion section that this case is only illustrative, and that additional cases will be required for statistical assessment of ecotypes, including the spatial organization of HB cell states and their interaction with their microenvironment.

3. Fig. 1c visualization and insight are confused. Why would the author show the batches in the UMAP, and what information can be derived from this UMAP? Usually, the UMAP should visualize the post-integration cell cluster and show the consistent cell type across multiple conditions.

Contrary to normal cells that are expected to display homogeneous transcriptomes in different samples, tumor cells have important sources of inter-sample variability. Each tumor has its own set of driver alterations. In particular, copy-number changes lead to the over / underexpression of hundreds of genes located on gained / lost chromosomes. As a result, tumor cells usually cluster by sample in UMAPs, contrary to normal cells from the microenvironment that cluster by cell type. This is not the result of a batch effect but true biological differences. Integration methods like Harmony or scVi are designed to integrate cells with similar transcriptomes across samples, which is not the case of tumor cells. Instead, several groups have used principal component analysis (Tirosh, Science 2016), or other dimension reduction techniques like multidimensional scaling (Patel, Science 2014) or non-negative matrix factorization (Kinker, Nat Genet 2020; Gavish, Nature 2023), to overcome the inherent inter-sample variability of tumor cells and identify shared transcriptional programmes. This is the strategy we used in this work.

Importantly, our 6 HB samples were processed the same day, with the same protocol and by the same engineer to reduce experimental batch effects. We also verified that non-tumor cells cluster by cell type and not by sample of origin, precluding the presence of major batch effects in our data. In the revision, we have explained why we did not apply integration methods (Results paragraph "Hepatoblastoma cells display continuous states along two differentiation axes"), and we have modified **Fig. 1c** (see below) to clearly show that non-tumor cells cluster by cell type (in both snATAC-seq and snRNA-seq UMAPs) and only tumor cells cluster by sample, in agreement with their tumor-specific copy-number changes (**Fig. 1f**).

Revised Fig. 1c. Uniform manifold approximation and projection (UMAP) of all cells from the 6 HB based on their snATAC-seq (left) and snRNA-seq (right) profiles, annotated by sample of origin (top) or cell type (bottom).

4. The authors likely overinterpret the PCA findings in Fig. 2. There are comments on PCA revealing two main axes of variation, yet no scree plot is provided to support such a claim. While PC1 and PC2 will certainly contain more variation than PC3 and beyond, most scRNA-Seq datasets will appropriately contain the most variance in at least 10 PCs. The only differentiable axis in Fig. 2A is PC1, but the dataset only includes one mesenchymal sample, so there is no way of knowing if this is an outlier result or a real distinct population.

As suggested, we have now provided the scree plot (below) in **Fig.2a**. Our analysis of Song's scRNA-seq data (see answer to comment #1) also retrieved 2 main PCs related to the same differentiation states, supporting the robustness of these 2 differentiation axes in HB. Of note, PC1 and 2 are used in this study to classify cells by differentiation state, but their full transcriptome and chromatin accessibility profiles are then used to define the gene regulatory networks and master TFs. However, we fully agree that other PCs (in particular PC3-10, as anticipated by the Reviewer) also account for a substantial part of the variance and certainly contain valuable biological information. This work focuses on differentiation states, but future analyses will be useful to extract the exhaustive catalog of transcriptional modules in HB. This point is now mentioned in the **Discussion** section.

Revised Fig. 2a,b. **a** Screeplot showing the standard deviation of principal components (PCs) in the snRNA-seq data of tumor cells. **b** Projection of tumor cells from the 6 HB samples on the 2 first principal components. Tumor cells are colored by sample. Density plots show the distribution of cells from each sample along the two PCA axes.

Regarding PC1, we agree that it was supported by the single mesenchymal sample, and that extra evidence was needed to confirm its link with the mesenchymal subtype (and not just sample #3610T). A reassuring finding was the fact that the GRN associated with this population comprised several mesenchymal transcription factors (TWIST1/2, ZIC1/4, TBX4...). In the revised manuscript, we used Bisque deconvolution tool to quantify our HB cell states in a compendium of 314 bulk RNA-seq samples from 5 studies, and we validated the association of the scM signature with mesenchymal histology and molecular group (see response to comment #1).

5. This manuscript could have really benefitted from an external look at bulk RNA-Seq or ATAC-Seq data to identify if the trends seen in the limited-sample single-cell data are present in a larger sample. The authors attempted this on a small cohort but excluded mesenchymal samples, which exist in the center of the scLP/scH continuum in PC2.

We fully agree. In the revision, we have validated our findings using extensive published single-cell RNA-seq (6,244 cells from 9 patients, Song, Nat Commun 2022) and bulk RNA-seq data (314 HB samples from 5 series: Carrillo-Reixach, JHEP2020; Hooks, Hepatology 2018; Sekiguchi, NPJ 2020; Nagae, Nat Commun 2021, Hirsch, Cancer Discov 2021). These new data are described in response to comment #1.

We did not find published HB ATAC-seq data set, but we used ChIP-seq data from the ReMap database (<https://remap2022.univ-amu.fr>) to provide additional evidence supporting the master TFs of each cell state. The 4th release of ReMap (2022) compiles 8,103 quality controlled ChIP-seq datasets from various public sources such as GEO, ArrayExpress and ENCODE. We systematically analyzed the overlap between ATAC-seq peaks more accessible in each cell state and TF ChIP-seq peaks from ReMap. Although several TFs were not represented in ReMap (or not in a relevant cell type), we found a significant association for 6/14 TFs of the scH module (AR, CEBPB, CEBPD, POU5F1, HLF and NIL3), 7/14 TFs of the sc-epi module (HNF1A, HNF4A, HNF4G, ONECUT1, ONECUT2, FOXA1 and GATA4), 4/11 TFs of the scLP module (MAZ, MYCN, FOXA2 and BCL11A) and 4/22 TFs of the scM module (LEF1, RUNX2, TCF3 and ESR1). These TFs with ChIP-seq validation are now indicated in **Sup Table 7**. Representative TF-binding regions (see below) are shown in **Fig. 4d** and **Sup Fig. 11**.

d

New Fig. 4d showing representative examples of overlaps between ATAC-seq and ChIP-seq peaks of TFs associated with HB cell states (HNF1A for sc-epi, LEF1 for scM, CEBPB for scH and MAZ for scLP).

CEBPB (scH GRN)

Overlap of CEBPB ChIP-seq peaks with the 5,193 ATAC-seq peaks more accessible in scH: $q=4.5e-6$

MAZ (scLP GRN)

Overlap of MAZ ChIP-seq peaks with the 9,432 ATAC-seq peaks more accessible in scLP: $q=1.4e-5$

HNF1A (sc-epi GRN)

Overlap of HNF1A ChIP-seq peaks with the 28,836 ATAC-seq peaks more accessible in sc-epithelial: $q=3.0e-122$

LEF1 (scM GRN)

Overlap of LEF1 ChIP-seq peaks with the 20,010 ATAC-seq peaks more accessible in scM: $q=1.9e-15$

New Sup Fig. 11. Overlap between differentially accessible ATAC-seq peaks and transcription factor ChIP-seq peaks. Eight representative regions are shown, displaying the overlap between ChIP-seq peaks of 4 key transcription factors (HNF1A associated with sc-epithelial cells, LEF1 with scM cells, CEBPB with scH cells and MAZ with scLP cells), together with the normalized ATAC-seq signals in scH, scLP and scM cells. All significant associations are reported in **Sup Table 7**. ChIP-seq coverage tracks were obtained from ENCODE (HNF1A in HepG2 cells: ENCFF502ACF; CEBPB in HepG2 cells: ENCFF406BBU; MAZ in HepG2 cells: ENCFF527EZL) or GEO (LEF1 in hESC: GSM1579343).

6. Regarding cell population identification, how did the author solve the integration issue among multiple samples? I cannot find the description for the integration analysis. PCA cannot solve the integration and identify consistent cell types based on multiple samples. I highly recommend that the author redo this analysis using integration tools like Seurat, Harmony, or scVI.

As explained in the reply to point #3, integration methods like Harmony or scVi are designed to integrate cells with similar transcriptomes across samples, which is not the case of tumor cells. Forcing the integration of tumor cells would result in the correction of true biological variation, including genes that are up / down-regulated by tumor-specific copy-number changes. Principal component analysis (PCA)

was previously used to separate biological from technical and tumor-specific variations in scRNA-seq data, and to identify shared transcriptional programmes in tumor cells (Tirosh, Science 2016). We applied a similar strategy in this work, after controlling that non-tumor cells clustered by cell type and not by sample in our dataset (revised **Fig. 1c**). We have clarified this point in the revision (**Results** paragraph "Hepatoblastoma cells display continuous states along two differentiation axes") and explained why we did not apply integration methods.

7. For the spatial analysis, the deconvolution analysis (e.g., using cell2location) is necessary to investigate cell composition in three regions, including fetal, embryonal, and intersection regions. As suggested, we used a computational tool (SPOTlight, Elosua-Bayes *et al.*, NAR 2021) to deconvolute HB cell states in Visium spots. Cell composition was associated with histological annotations, with an enrichment of scLP cells in embryonal spots and scH cells in fetal spots. scH/LP cells were quite abundant, although this population is heterogeneous and may include a range of phenotypes between scLP and scH (see answer to your minor comment #2). We have added this analysis in the Results section and in the new **Fig. 3b** and **Supplementary Fig. 8** (see below).

Revised Fig. 3a,b. Spatial transcriptomics analysis of tumor #3133T. a Hematoxylin and eosin staining in an FFPE section of tumor #3133T. The two annotated regions correspond to embryonal nodules surrounded by fetal areas. **b** Ternary graph showing the deconvoluted proportions of epithelial HB cell states in each Visium spot of regions 1 (left) and 2 (right). Visium spots were annotated by a pathologist and are colored by histology.

New Supplementary Fig. 8. Deconvolution of HB cell states in spatial transcriptomics data. a Deconvolution of HB cell states in region 1. The upper left panel shows the histological annotations of spots. The upper right panel shows a zoomed-in focus on H&E staining of a region at the interface between fetal and embryonal cells. The bottom panel shows the results of the deconvolution, with a pie chart showing the estimated cell state proportions in each spot. b Deconvolution of HB cell states in region 2.

8. When selecting the TF target modules in Fig. 5, how do you justify selecting four modules naturally? The dendrogram at the top shows there's more variability within the scLP targets than there is between sc-epi and scH targets. It can be quite convenient to see roughly the results you're looking for, but if you split one with less variation, then you need to split the other. This is pretty clear in the Fig. 5b plots where scH/LP and scH do not really differentiate much from each other.

We thank the Reviewer for this important remark. Fig. 5 presents a bi-clustering of TFs (rows) and their targets (columns). The 4 modules were defined naturally based on the dendrogram of the TFs (rows). As noted by the Reviewer, the sc-epi and scH modules are quite close. However, their activation pattern is slightly different: the sc-epi module is expressed similarly in scLP, scH/LP and scH cells, whereas the expression of the scH module increases gradually. The latter module seems associated with the most terminal step of hepatocytic differentiation (**Fig. 6**). In the revision, we have validated the TF-target modules in 6 independent data sets (see answer to comment #1). We have also clarified the fact that they were defined based on the TF clustering in the Methods section:

"The clustering of the TFs revealed 4 modules (Fig. 5a), which we named scH, scLP, scM and sc-epi according to their level of activation in each cell state. The sc-epi and scH modules were quite close in terms of target genes, but they displayed different dynamics of activation between scLP and scH states (Fig. 5b) so we decided to keep them separated."

Minor Comments:

1. For the conclusion of Supplementary Fig. 3, please provide a quantitative description to showcase the consistency between WGS data and single-cell inferred copy number profiles (such as correlation).

As suggested, we generated 2 additional panels for Supplementary Fig. 3 in the revised manuscript (see below):

- panel g shows the distribution of InferCNV expression ratios as a function of the copy-number status of the gene in WGS data

- panel h shows the correlation between InferCNV expression ratio and the WGS coverage log-ratio at the gene location.

Both representations illustrate the high consistency between InferCNV and WGS copy-number data.

g**h**
New Supplementary Fig. 5g-h: Correlation between InferCNV and WGS copy-number data. **g**, Boxplots showing the distribution of InferCNV expression ratios as a function of the segment's copy-number status derived from WGS data. Note that deletions were only detected in sample #3133T. **h**, Correlation between InferCNV expression ratios and WGS coverage log-ratio at the gene position.

2. There is very little justification for the four populations in Fig. 2. The gene sets clearly denote the three cell states previously identified. Beyond Fig. 2, the authors do not utilize this hybrid group much, nor do they explain them any further.

We defined scH, scLP and intermediate scH/LP cells using fixed thresholds on PC1. This is convenient to identify and name intermediate cells, in order to explore their chromatin accessibility profiles (**Fig. 2d**), and their distribution in UMAPs (**Fig. 2f,g**). However, we fully agree that epithelial cells (scH, scLP and scH/LP) define a continuum of differentiation rather than separate groups. We have now clarified this point in the **Discussion** section.

3. In Fig. 3, it would have been helpful to show scH and scLP markers on the zoomed-in region 1 and 2 to confirm the conclusions. It looks as though these markers are not that great at differentiating pathologist definitions on one of the two axes, which weakens the statements made. This is especially true for region 2, where scH markers essentially cannot differentiate between Embryonal and Fetal populations.

We thank the Reviewer for this remark. We have added the expression of scH and scLP markers in the zoomed-in regions 1 and 2 in the revised **Fig. 3** (see below). Regarding the fetal spot with low expression of scH markers in region 2, we noticed that these spots were located at the border of the slide, in an area where the tissue was folded (see below). We have now removed the spots from this folded area, and the association of scH markers with fetal spots is much clearer.

Correction of spot annotations for a zone in region 2 containing folded tissue.

Revised Fig. 3. Spatial transcriptomics analysis of tumor #3133T. *a* Hematoxylin and eosin staining in an FFPE section of tumor #3133T. The two annotated regions correspond to embryonal nodules surrounded by fetal areas. *b* Ternary graph showing the deconvoluted proportions of epithelial HB cell states in each Visium spot of regions 1 (left) and 2 (right). Visium spots were annotated by a pathologist and are colored by histology. *c* Focus on region 1 of the FFPE slide with a nodule of embryonal cells surrounded by fetal cells. *d* Spatial distribution of the mean expression of scH and scLP markers (Supplementary Table 3) in region 1. *e* Anti-correlation of scH and scLP markers across Visium spots in region 1, colored according to their histological annotation in panel *c*. *f* Focus on region 2 of the FFPE slide, with spots annotated by histology. *g* Spatial distribution of the mean expression of scH and scLP markers in region 2. *h* Anti-correlation of scH and scLP markers across Visium spots in region 2, colored according to their histological annotation in panel *f*.

4. In the result, section of “Hepatoblastoma cells display continuous states along two differentiation axes,” UMAP and t-SNE is not cell cluster identification method. It only provides visualization for cellular homogeneity and heterogeneity.

We agree and we have rephrased as follows: "Uniform Manifold Approximation and Projection (UMAP) and t-distributed stochastic neighbor embedding (t-SNE) are widely used to explore cellular heterogeneity in single-cell data. Yet, when applied to cancer data, both methods tend to be highly sensitive to copy-number and group tumor cells by sample of origin rather than molecular group."

5. What is the mosaic cell? Please define it.

The term "mosaic cell" was indeed ambiguous. It is not the cells that are mosaic but the alteration of 11p15 locus, that is present in only a fraction of normal liver liver cells. We have thus rephrased as follows: "We also analyzed 2 matched non-tumor liver samples, one of which (#3377N) showed pre-neoplastic colonization by cells with mosaic alteration of 11p15 locus⁵."

6. Provide a detailed figure legend in Fig. 1c.

We have now modified **Fig.1c** to show the cell type annotation of tumor and non-tumor cells, and we provided a detailed figure legend as requested.

Reviewer #4 (Remarks to the Author): Expert in single-cell multi-omics, spatial transcriptomics, gene regulatory networks, cancer genomics, and bioinformatics

Overview:

The manuscript (ms) presents a comprehensive analysis of hepatoblastoma cell transcriptomes, epigenomic landscapes, and somatic alterations at the single-cell level. The study successfully identifies differentiation between hepatoblastoma cell states (Hepatocytic, Liver Progenitor, H/LP intermediate, and Mesenchymal) and correlates them with prenatal liver development signatures. Additionally, the investigation of epigenetic remodeling, gene regulatory networks, and the interplay of clonal evolution and phenotypic plasticity provides valuable insights into hepatoblastoma progression and response to chemotherapy. Overall, this ms is well-written, the comprehensive data analysis results are well-presented, and this study significantly contributes to our understanding of hepatoblastoma biology. Nevertheless, I have a few comments that, if addressed, would significantly enhance the strength of the ms.

We warmly thank the Reviewer for his/her positive assessment of our work and for the constructive comments that helped us enhance the strength of the manuscript.

Comments:

1. In this study, it is mentioned that six hepatoblastoma (HB) samples were selected from the Hirsch data set, which contains 100 bulk RNA-seq samples. It would be helpful if the authors could provide more information on how these specific six samples were chosen. Was the selection random, or were there specific criteria used to ensure representativeness of the samples? Clarifying the sample selection process would enhance the transparency and reproducibility of the study.

The 6 samples included in this study were carefully selected to account for the molecular diversity of HB. We used matched bulk RNA-seq to ensure that all molecular groups were represented. This was shown initially by principal component analysis (Fig. 1a). In the revised manuscript, we also annotated these samples in the hierarchical clustering of 100 HB from Hirsch series, showing that they represent the diversity in the hierarchical clustering of molecular subgroups: 3 samples belong to the 'Liver Progenitor', 2 to the 'Hepatocytic' and 1 to the 'Mesenchymal' subgroup (new **Sup Fig. 1**, see below).

Unsupervised clustering of 100 HB and 4 nontumor liver samples (bulk RNA-seq, Hirsch 2021 series)

New Supplementary Fig. 1. Transcriptomic classification of the 6 samples selected for single-nucleus Multiome. The dendrogram represents the hierarchical clustering of bulk RNA-seq profiles of 100 HB

and 4 non-tumor liver samples (Hirsch data set). The 6 samples selected for snMultiome are indicated below, together with bulk molecular groups from the original study and sample type (pre / post-chemotherapy).

In addition, we have added in **Sup Fig. 2** the pathological reviewing of the mirror blocks, indicating the proportions of fetal, embryonal and mesenchymal cells. Some samples were pure (e.g. #3610T, 100% mesenchymal) whereas others contained mixtures of histological components (e.g. #2959T, 70% embryonal and 30% fetal). For one patient, we selected two regions: one with a dominance of embryonal cells classified 'Liver Progenitor' in bulk RNA-seq, the other with a dominance of fetal cells classified 'Hepatocytic' in bulk RNA-seq.

Overall, this selection allowed us to cover the molecular and histological diversity of HB, and to investigate both inter- and intra-sample heterogeneity. In the revised manuscript, we have clarified the selection process in the first paragraph of the **Results** section.

2. Given the high variability and heterogeneity of human samples, it is crucial to address the potential batch (patient) effect when presenting the six snRNA-seq samples together on UMAP. However, the manuscript does not mention any data integration step or correction for the patient-specific effect. It would be valuable if the authors could discuss how they accounted for this strong patient-specific effect in the analysis. Describing any normalization or batch correction methods employed would strengthen the robustness of the results and ensure that the observed patterns are not solely driven by patient-specific differences.

Contrary to normal cells that are expected to display homogeneous transcriptomes in different samples, tumor cells have important sources of inter-sample variability. Each tumor has its own set of driver alterations. In particular, copy-number changes lead to the over / underexpression of hundreds of genes located on gained / lost chromosomes. As a result, tumor cells usually cluster by sample in UMAPs, contrary to normal cells from the microenvironment that cluster by cell type. This is not the result of a batch effect but true biological differences. Integration methods (like Harmony or scVi) are designed to integrate cells with similar transcriptomes across samples, which is not the case of tumor cells. Forcing the integration of tumor cells would result in the correction of true biological variation, including genes that are up / down-regulated by tumor-specific copy-number changes. Instead, several groups have used principal component analysis (Tirosh, Science 2016), or other dimension reduction techniques like multidimensional scaling (Patel, Science 2014) or non-negative matrix factorization (Kinker, Nat Genet 2020; Gavish, Nature 2023), to overcome the inherent inter-sample variability of tumor cells and identify shared transcriptional programmes. This is the strategy we used in this work.

Importantly, our 6 HB samples were processed the same day, with the same protocol and by the same engineer to reduce experimental batch effects. We also verified that non-tumor cells cluster by cell type and not by sample of origin, precluding the presence of major batch effects in our data. In the revision, we have explained why we did not apply integration methods (**Results** paragraph "Hepatoblastoma cells display continuous states along two differentiation axes"), and we have modified **Fig. 1c** (see below) to clearly show that non-tumor cells cluster by cell type (in both snATAC-seq and snRNA-seq UMAPs) and only tumor cells cluster by sample, in agreement with their tumor-specific copy-number changes (**Fig. 1f**).

Revised Fig. 1c. Uniform manifold approximation and projection (UMAP) of all cells from the 6 HB based on their snATAC-seq (left) and snRNA-seq (right) profiles, annotated by sample of origin (top) or cell type (bottom).

3. In the visium spatial data analysis, it is stated that spots containing a mixture of tumor and stromal cells or hematopoiesis were excluded. It would be helpful if the authors could elaborate on how the cell type mixture was identified and distinguished from pure tumor or stromal cells. Providing details about the criteria or markers used to determine the cell type composition of each spot would enhance the clarity and reliability of the spatial analysis results.

Spots containing a mixture of tumor and stromal cells were manually identified by an expert pathologist based on the examination of H&E images in 10X Genomics *Loupe Browser* tool. We have clarified this point in the **Methods** section.

In addition, as suggested by Reviewer #3, we used a computational tool (SPOTlight, Elosua-Bayes *et al.*, NAR 2021) to deconvolute HB cell states in Visium spots. Cell composition was associated with histological annotations, with an enrichment of scLP cells in embryonal spots and scH cells in fetal spots. ScH/LP cells were quite abundant, although this population is heterogeneous and may include a range of phenotypes between scLP and scH. We have added this analysis in the Results section and in the new **Fig. 3b** and **Supplementary Fig. 8** (see below).

Revised Fig. 3a,b. Spatial transcriptomics analysis of tumor #3133T. a Hematoxylin and eosin staining in an FFPE section of tumor #3133T. The two annotated regions correspond to embryonal nodules surrounded by fetal areas. **b** Ternary graph showing the deconvoluted proportions of epithelial HB cell states in each Visium spot of regions 1 (left) and 2 (right). Visium spots were annotated by a pathologist and are colored by histology.

New Supplementary Fig. 8. Deconvolution of HB cell states in spatial transcriptomics data. a Deconvolution of HB cell states in region 1. The upper left panel shows the histological annotations of spots. The upper right panel shows a zoomed-in focus on H&E staining of a region at the interface between fetal and embryonal cells. The bottom panel shows the results of the deconvolution, with a pie chart showing the estimated cell state proportions in each spot. *b* Deconvolution of HB cell states in region 2.

REVIEWER COMMENTS

Reviewer #2 (Remarks to the Author):

The authors carefully addressed all my questions, added data and modified figures accordingly. I have no further comments.

EDITORIAL NOTE

Reviewer #2 considers that the following comments from Reviewer #1's original report require additional work:

* The Reviewer considers that, while the addition of publicly available HB data is commendable, the conclusions remained descriptive; in Figure 1, cancer cells clustered in isolation and it is unclear if previously known subtypes were identified. Please discuss related limitations as well as potential reasons, such as sample size and currently available technologies.

* The TF analysis was improved but it should be demonstrated whether claims are predictive; this could be done with simple statistical analyses. For example: please show whether CEBPB sites in the ReMap database are significantly more enriched in scH open regions than in other regions - this could be done with an F test; please also show that the binding sites for these TFs are amongst the most enriched in the ReMap database.

Reviewer #3 (Remarks to the Author):

Response: I agree with the author's claim that tumor cells from multi-samples are hard to distinguish between biological variance and technical variance. However, from the fig. 1c, although different non-tumor cell types (e.g., macrophage, T cell, endothelial, and liver stellate) can be grouped separately, I can tell the cells within one cluster were still grouped by individual patients. This may indicate differences between individual patients existed. Therefore, evaluating the variance of biological differences and individuals may help to strengthen the findings in heterogeneous tumors. For example, the variance can be estimated by using PC regression (Malte D. Luecken, Nature Methods, 2022).

Reviewer #4 (Remarks to the Author):

I appreciate the authors' diligent efforts in addressing the comments and concerns I raised regarding the bioinformatics analysis.

In the updated manuscript, they have incorporated additional information about the sample selection process. They have also provided a clear justification for their choice of the current dimension reduction approach. The inclusion of the deconvolution analysis has further enhanced the understanding of the Visium data analysis.

Overall, I find the revised manuscript to be satisfactory.

Reviewer #2 (Remarks to the Author):

The authors carefully addressed all my questions, added data and modified figures accordingly. I have no further comments.

We warmly thank the Reviewer for his/her positive assessment of our revised manuscript.

Reviewer #2 considers that the following comments from Reviewer #1's original report require additional work:

* The Reviewer considers that, while the addition of publicly available HB data is commendable, the conclusions remained descriptive; in Figure 1, cancer cells clustered in isolation and it is unclear if previously known subtypes were identified. Please discuss related limitations as well as potential reasons, such as sample size and currently available technologies.

Cancer cells indeed cluster by sample in the UMAPs of Figure 1, as in most cancer single-cell data sets. As explained in the revised manuscript, tumor cells display important inter-sample biological variation, notably due to tumor-specific driver alterations and copy-number changes, hence usually cluster separately in UMAPs. This is why we used principal component analysis to identify shared transcriptional programmes, as previously described (Tirosh, Science 2016). This point is explained in the Results section of the revised manuscript:

"Uniform Manifold Approximation and Projection (UMAP) and t-distributed stochastic neighbor embedding (t-SNE) are widely used to explore cellular heterogeneity in single-cell data. Yet, when applied to cancer data, both methods tend to be highly sensitive to copy-number and group tumor cells by sample of origin rather than molecular group. This was also the case in our series (Fig. 1c). Integration methods (like Harmony¹⁴ or scVI¹⁵) are well suited to integrate normal cells, expected to have similar transcriptomes across samples. By contrast, tumor cells display important inter-sample biological variation, notably due to tumor-specific driver alterations and copy-number changes. Principal component analysis (PCA) was previously used to separate biological from technical and tumor-specific variations in scRNA-seq data, and to identify shared transcriptional programmes in tumor cells¹⁶. Applied to our 14,448 tumor cells, PCA revealed 2 main axes of variation..."

Regarding the **novelty of the subtypes**, we had previously annotated cells in a single patient using signatures from bulk RNA-seq groups to illustrate intra-tumor heterogeneity (Hirsch, Cancer Discov 2021). However, the definition of single-cell HB subtypes is new to this study. By analyzing 14,448 tumor cells belonging to 6 samples representative of HB diversity, we could reveal a continuum of differentiation states in HB, and an intermediate scH/LP state between scH and scLP poles. The comparison of our HB cell states with previous bulk and single-cell HB signatures, and the added value of this study, are discussed in the Discussion section of the revised manuscript:

"[Our HB cell states] show similarities with HB cell subtypes identified by Huang et al. (1,293 tumor cells from 5 HB patients)¹² and HB tumor signatures identified by Song et al. (6,244 tumor cells from 9 HB patients)¹³. More precisely, our scM state is similar to Huang's HB2 (mesenchyme-like) group and expresses Song's 'fibroblast-like' signature. Our scLP subtype is closer to Huang's HB1 (progenitor-like) group, and our scH state has the highest expression of Huang's HB3 (hepatocyte-like) and Song's 'Hepatoblast' markers (Supplementary Fig. 16). However, by analyzing a larger number of cells (14,448 tumor cells in our study) from samples representative of the main bulk molecular groups, and applying a dimension reduction technique less sensitive to tumor-specific effects, we unraveled a continuum of states between scLP, scH and scM cells. In particular, we defined an intermediate population of scH/LP cells expressing both H and LP markers at low levels, which we validated in spatial transcriptomics. ScH/LP cells do not correspond to a separate population, but they display a continuum of differentiation states between scLP and scH."

As requested, we have added in the Discussion section a paragraph indicating the **limitations of the study**, notably regarding sample size and currently available technologies:

"A limitation of this study is the small sample size. Single-cell multiomic analyses of larger series may reveal additional HB cell states not represented in the current data set. In addition, ATAC-seq only gives access to chromatin accessibility profiles. Profiling other epigenetic features, like DNA methylation³⁶ or histone modifications³⁷, will be useful to understand how the different layers of epigenetic regulation are orchestrated and their relative roles in HB cell plasticity."

* The TF analysis was improved but it should be demonstrated whether claims are predictive; this could be done with simple statistical analyses. For example: please show whether CEBPB sites in the ReMap database are significantly more enriched in scH open regions than in other regions - this could be done with an F test; please also show that the binding sites for these TFs are amongst the most enriched in the ReMap database.

We have indeed analyzed the overlap between ChIP-seq peaks in ReMAP database and differentially accessible ATAC-seq peaks for each cell state using the *ReMapEnrich* R package. These statistics are shown in **Supplementary Table 7** and for representative TFs (CEBPB, MAZ, HNF1A and LEF1) in **Sup Fig. 11**. Regarding CEBPB for example, we found a significant overlap between CEBPB ChIP-seq peaks in ReMAP and the 5,193 ATAC-seq peaks more accessible in scH cells ($q=4.5e-6$). We have now added these statistics also in the legend of Fig. 4d to clarify this point:

"**Fig. 4d** Significant overlap between differentially accessible ATAC-seq peaks and transcription factor ChIP-seq peaks from ReMAP database. The 4 examples shown include HNF1A associated with sc-epithelial cells (overlap between HNF1A ChIP-seq peaks and ATAC-seq peaks more accessible in sc-epithelial cells: $q=3.0e-122$), LEF1 with scM cells ($q=1.9e-15$), CEBPB with scH cells ($q=4.5e-6$) and MAZ with scLP cells ($q=1.4e-5$). Additional examples are shown in **Supplementary Fig. 11**, and all significant associations are reported in **Supplementary Table 7**. ChIP-seq coverage tracks were obtained from ENCODE (HNF1A in HepG2 cells: ENCFF502ACF; CEBPB in HepG2 cells: ENCFF406BBU; MAZ in HepG2 cells: ENCFF527EZL) or GEO (LEF1 in hESC: GSM1579343)."

As suggested, we have also verified that the binding sites for TFs belonging to the GRNs of each cell state were among the most enriched in the ReMAP database (see Figure below). We found a significant enrichment of GRN TFs among the top-ranked TFs of ReMAP database, with p-values $<8.4e-5$. Of note, other TFs display highly significant association between their ReMAP ChIP-seq peaks and differentially accessible peaks in cell states, but they were not included in the GRNs because their composite scores (considering snATAC-seq motif deviation as well as bulk and single-cell expression changes) were not in the top 20 (see Methods).

These figures have been added in **Sup Fig. 11b**.

Supplementary Figure 11. Overlap between differentially accessible ATAC-seq peaks and transcription factor ChIP-seq peaks. b Distribution of GRN TFs among ReMAP ChIP-seq peaks enriched in the differential ATAC-seq peaks of each cell state.

Reviewer #3 (Remarks to the Author):

Response: I agree with the author's claim that tumor cells from multi-samples are hard to distinguish between biological variance and technical variance. However, from the fig. 1c, although different non-tumor cell types (e.g., macrophage, T cell, endothelial, and liver stellate) can be grouped separately, I can tell the cells within one cluster were still grouped by individual patients. This may indicate differences between individual patients existed. Therefore, evaluating the variance of biological differences and individuals may help to strengthen the findings in heterogeneous tumors. For example, the variance can be estimated by using PC regression (Malte D. Luecken, Nature Methods, 2022).

We warmly thank the Reviewer for this insightful comment.

First, it is important to note that hepatoblastomas (HB) differ not only by the phenotype of their tumor cells, but also by the abundance, composition and functional orientation of their immune infiltrates. For example, we previously showed that APC-mutated HB have massive intratumor tertiary lymphoid structures (Morcrette *et al.*, Oncoimmunology 2019) and that liver progenitor HB samples are "immune-cold" compared to hepatocytic HB samples (Hirsch *et al.*, Cancer Discov 2021). We thus expect that some of the patient-related variance of non-tumor cells will also be of biological origin.

As suggested, we used PC regression to estimate the proportion of variance explained by cell type and sample of origin in non-tumor cells. We focused on the first 5 principal components (PC1-5) that together account for 61.5% of the variance in non-tumor cells (see scree plot below).

Proportion of variance explained by each principal component in non-tumor cells

We first used linear regression (*lm* function in R statistical software) to estimate the proportion of each PC explained by cell type: $R2(PCi, cell\ type)$. We then estimated the proportion of PC1-5 variance explained by cell type as the sum of $R2$ values, weighted by the variance of each PC, divided by the total variance of PC1-5:

$$\frac{\sum_{i=1}^5 R2(PCi, cell\ type) * var(PCi)}{\sum_{i=1}^5 var(PCi)} = 0.686$$

Thus, cell type explains 68.6% of the variance of the first 5 PCs.

Next, we repeated the regression analysis including both cell type and sample of origin as explanatory variables. We estimated the proportion of each PC explained by both factors: $R2(PCi, cell\ type \ \& \ sample\ of\ origin)$. Finally, we calculated the proportion of PC1-5 variance explained by cell type and sample of origin:

$$\frac{\sum_{i=1}^5 R2(PCi, \text{cell type \& sample of origin}) * var(PCi)}{\sum_{i=1}^5 var(PCi)} = 0.828$$

Thus, cell type and sample of origin explain 82.8% of the variance of the first 5 PCs. In other words, sample of origin explains 14.1% of the variance in addition to the 68.6% explained by cell type. This extra 14.1% of explained variance may be due to technical reasons (batch effect) and/or biological differences between the immune infiltrates of the samples.

We have added this analysis in the new "**Batch effect estimation**" paragraph of the **revised Methods section**.

Reviewer #4 (Remarks to the Author):

I appreciate the authors' diligent efforts in addressing the comments and concerns I raised regarding the bioinformatics analysis.

In the updated manuscript, they have incorporated additional information about the sample selection process. They have also provided a clear justification for their choice of the current dimension reduction approach. The inclusion of the deconvolution analysis has further enhanced the understanding of the Visium data analysis.

Overall, I find the revised manuscript to be satisfactory.

We warmly thank the Reviewer for his/her positive assessment of our revised manuscript.

REVIEWERS' COMMENTS

Reviewer #2 (Remarks to the Author):

The authors carefully addressed all my questions, and I recommend acceptance at this time.

Reviewer #3 (Remarks to the Author):

The Authors have addressed all my concerns.